

# The braincase anatomy of *Simosaurus gaillardoti* (Diapsida: Sauropterygia) revealed with X-ray micro-computed tomography

Elisa H. London[1], Dennis F.A.E. Voeten[2,3], Henning Blom[4] and Torsten M. Scheyer[5]

[1] Department of Earth Sciences, Uppsala Universitet, Uppsala, Sweden
[2] Frisian Museum of Natural History, Leeuwarden, Netherlands
[3] Vertebrate Evolution, Development and Ecology group, Naturalis Biodiversity Center, Leiden, Netherlands
[4] Department of Organismal Biology, Uppsala Universitet, Uppsala, Sweden
[5] Department of Paleontology, University of Zurich, Zurich, Switzerland

Corresponding author
Torsten M. Scheyer,
tscheyer@pim.uzh.ch

## ABSTRACT

Sauropterygia is a clade of Mesozoic marine reptiles that includes the eosauropterygian *Simosaurus gaillardoti* Von Meyer, 1842, classically considered to be a member of Nothosauroidea. The braincase of this species has thus far only been studied in acid-prepared specimens. Acid preparation is a destructive technique prone to information loss, *e.g.*, through the dissolution of thin braincase bones. Here, one well-preserved skull (SMNS 16363) that remains partially embedded in matrix has been visualised using X-ray micro-computed tomography, and the braincase region has been virtually extracted. This braincase provides valuable information on the general shape of the endocast, the existence and shape of epipterygoids, which were previously considered absent in the taxon, the course of cranial nerves and the bifurcation of the internal carotid arteries along an expanded and broad parabasisphenoid, the latter extending in a tapering cultriform process to the level of the external and internal narial openings. The arrangement of the semicircular canals of the inner ear confirms previously hypothesised adaptations for near-shore aquatic life in the species. The anatomical similarities of the braincases between *Simosaurus gaillardoti* and *Nothosaurus marchicus,* including a jugular foramen that is framed by the exoccipital medially and by the opisthotic laterally, support the current phylogenetic placement of the former as an early branching member of Nothosauroidea. The cranial flattening observed in nothosaurs relative to the less dorsoventrally flattened skull of *Simosaurus* reflects diverging feeding strategies. Most nothosaurs were fish-trap ambush predators, whereas *Simosaurus gaillardoti* had durophagous, as well as (opportunistic) piscivorous capacities. These results might indicate that specialised piscivorous predation using fish-trap dentition could be independently derived in nothosaurs and in pistosauroids (including plesiosaurs).

## INTRODUCTION

Sauropterygia is a diverse clade of marine reptiles that lived from the Lower Triassic until the K-Pg mass extinction (c. 252–66 Ma) (*Rieppel, 2000*; *Scheyer et al., 2014*; *Motani et al., 2017*). Although fossils are currently absent from the Latest Permian to the Earliest Triassic (*i.e.*, pre-Spathian), recent studies have discussed the possibility of an earlier appearance of Mesozoic marine reptile clades, including the sauropterygians, just prior to the Permian-Triassic Mass Extinction Event (PTME; *Wang et al., 2022*; *Kear et al., 2023*), in the case of *Wang et al. (2022)* based on tip-dating and clade-divergence timing estimates (these data were also re-used in *Kear et al., 2024*; but see also *Motani et al., 2017*). Sauropterygia is generally considered to include the two clades Placodontiformes and Eosauropterygia (Figs. 1A–1C), although there is an ongoing discussion on the internal topology of the phylogeny and the systematics are far from stable. Some studies divide Eosauropterygia into Pachypleurosauria, Pistosauroidea, and Nothosauroidea (*e.g.*, *Scheyer et al., 2017*; *Allemand, Moon & Voeten, 2023*). Saurosphargidae was recently added to the phylogenetic analyses as well, but its position as either within or closely related to Sauropterygia remains contested (*e.g.*, *Wang et al., 2022*; *Wolniewicz et al., 2023*).

Placodontiformes appeared in the early Anisian (247–242 Ma) in nearshore environments and Placodontia in particular exhibited a highly modified dentition specialised with flat and broad plate-like teeth facilitating their durophagous diet (*Rieppel, 2002*; *Neenan, Klein & Scheyer, 2013*; *Neenan et al., 2015*; *Pommery et al., 2021*; *Laboury et al., 2023*). Arguably the best-understood placodontoid genus, including the skull- and braincase anatomy, is *Placodus* (*Agassiz, 1833–45*), known from numerous isolated bones and teeth but also associated skeletons (*e.g.*, *Rieppel, 1995*; *Agassiz, 1833–45*; *Drevermann, 1933*; *Nosotti & Rieppel, 2002*; *Jiang et al., 2008*; *Neenan & Scheyer, 2012*; *Klein et al., 2022*). *Placodus* was a bottom feeder, mostly preying on shelled marine invertebrates (*Rieppel, 2002*; *Klein et al., 2015*; *Xing et al., 2020*). Among Eosauropterygia, Pachypleurosauria potentially appeared in the fossil record in the Olenekian (c. 251–247 Ma) with taxa such as *Majiashanosaurus discocoracoidis* (*Jiang et al., 2014*; *Wang et al., 2022*), but its placement among Sauropterygia has also been contested, as it was also found outside of Eosauropterygia (*Wolniewicz et al., 2023*), or as successive sister taxon (with *Hanosaurus*) to Nothosauroidea plus Pachypleurosauria (*Li & Liu, 2020*). Pachypleurosauria sensu stricto mostly fed on small, soft-bodied animals (*Rieppel, 2002*) and the majority of pachypleurosaurs are equipped with teeth consisting of a single morphology: small, conical teeth suited for processing soft-bodied invertebrates and fish (*Rieppel, 2002*). Facultatively durophagous forms are also known, evidenced by *Anarosaurus heterodontus* (*Klein, 2009*). Pistosauroidea, which in the Late Triassic gave rise to the globally distributed open-water plesiosaurs including the genus *Plesiosaurus*, were piscivorous (*Storrs, 1993*; *Wintrich et al., 2017*).

Nothosauroidea includes Olenekian forms that are among the oldest currently recognised sauropterygian fossils (*Li & Liu, 2020*; *Scheyer, Neuman & Brinkman, 2019*; *Kear et al., 2024*). Nothosaurs are mostly found in coastal or platform deposits and are considered

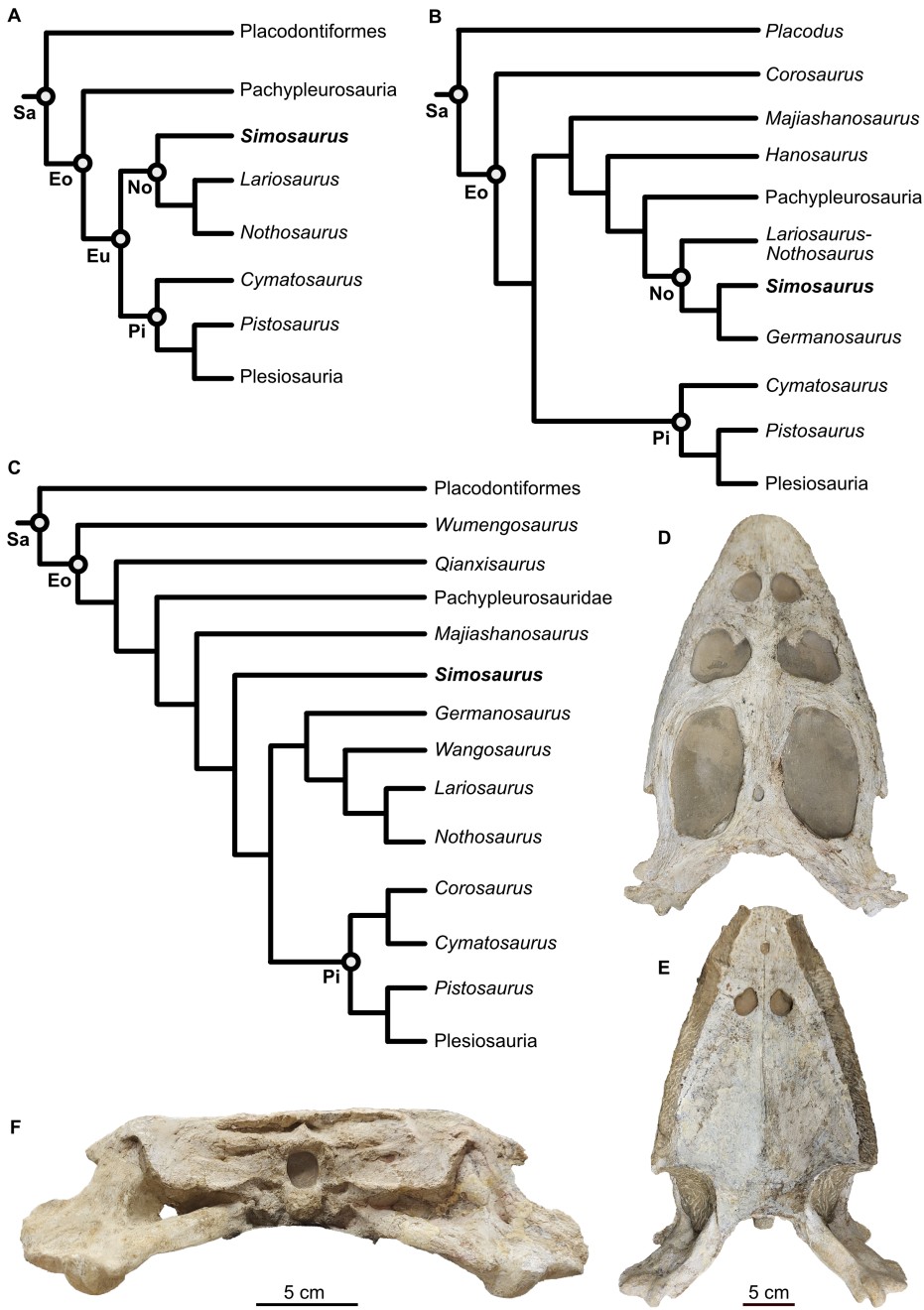

**Figure 1 Phylogenetic framework of Sauropterygia and studied specimen.** (A) Hypothesis modified from *Allemand, Moon & Voeten* (*2023* and references therein), reflecting the traditional topology of sauropterygians and *Simosaurus* as sister to other nothosaurs. (B) Hypothesis modified from *Li & Liu (2020)* in which *Simosaurus* was found being more highly nested within nothosaurs. (C) Hypothesis modified from *Wang et al. (2022)* showing *Simosaurus* as sister to the clade consisting of nothosaurs and pistosauroids. (D–F) Cranium of *Simosaurus gaillardoti* (SMNS 16363) in dorsal (D), ventral (E), and occipital (F) view. Abbreviations: Eo, Eosauropterygia; Eu, Eusauropterygia; No, Nothosauroidea; Pi, Pistosauroidea; Sa, Sauropterygia.

piscivorous but potentially include the more durophagous *Simosaurus* (*Storrs, 1993*; *Rieppel, 1999*; *Rieppel, 2002*; *Klein & Griebeler, 2018*).

The family Simosauridae contains two species in monospecific genera: *Simosaurus gaillardoti* (*Rieppel, 2000*; *De Miguel Chaves, Ortega & Pérez-García, 2018a*; Figs. 1D–1F) and *Paludidraco multidentatus* (*De Miguel Chaves, Ortega & Pérez-García, 2018b*). *P. multidentatus* is a relatively recently discovered simosaurid that was described based on one specimen found in the Keuper facies in central Spain dating from the Carnian to Norian (237–208 Ma) (*De Miguel Chaves, Ortega & Pérez-García, 2018b*). *S. gaillardoti* appeared in the Ladinian stage (242–237 Ma) and disappeared in the Carnian (237–227 Ma) (*Rieppel, 2000*; *Dalla Vecchia, 2008*; *Klein et al., 2016*; *De Miguel Chaves, Ortega & Pérez-García, 2018a*). The species can be found in Upper Muschelkalk and Lower Keuper stratigraphic units of mostly the western Tethyan region (*e.g.*, *Klein et al., 2016*) but has also been described in the Middle East (*Kear et al., 2010*; *Cabezuelo-Hernández, De Miguel Chaves & Pérez-García, 2024*). The phylogenetic position of *S. gaillardoti* remains debated, with most older studies classifying it as a nothosauroid (Fig. 1) despite distinct morphological differences from other nothosauroids (*Cheng et al., 2016*; *Klein et al., 2016*; *De Miguel Chaves, Ortega & Pérez-García, 2018a*; highly nested within Nothosauroidea as sister taxon to *Germanosaurus*: *Li & Liu, 2020*; *Hu, Li & Liu, 2024*). Other recent studies found Simosauridae (*i.e.*, *Simosaurus* and *Paludidraco*) as sister clade to Nothosauridae plus Pistosauroidea (*Wang et al., 2022*) or *Simosaurus* as the sister taxon to Pistosauridae (*Qiao et al., 2022*; *Wolniewicz et al., 2023*; note that other nothosaurs were not included in those studies, so a potential sister group relationship with any of those taxa was excluded from the analyses; see Figs. 1A–1C).

*Simosaurus gaillardoti* could reach 3–4 m in body length and possess flat and brevirostrine skulls, meaning they feature a broad jaw with a short rostrum (*Rieppel, 1994b*). *S. gaillardoti* crania have been shown to exhibit relatively large variation (*Rieppel, 1994a*; *De Miguel Chaves, Ortega & Pérez-García, 2018a*). This variation is, for example, expressed in the structural openings on the dorsal side of the skull (*De Miguel Chaves, Ortega & Pérez-García, 2018a*). Compared to other nothosaurs, *Simosaurus* has large, rounded upper temporal fenestrae (Figs. 1 and 2). Some nothosaurs have a specialised (mostly size-related) piercing dentition with large stout and often strongly recurved fangs followed by small, more gracile and often less-curved teeth (*e.g.*, *Rieppel, 2000*). This has been interpreted to facilitate ensnaring or trapping fish within the jaws, rather than piercing them (*Rieppel, 2002*; *Laboury et al., 2023*). *S. gaillardoti,* on the other hand, has teeth that are slightly bulbous and are more similarly sized (*Rieppel, 1994a*; *Rieppel, 2002*). This suggests a durophagous feeding strategy for *S. gaillardoti* (*Rieppel, 2002*). Based on its jaw mechanics, it might have been able to execute a strong and quick snapping bite to process ammonoids and hard-scaled fish (*Rieppel, 1994a*). The simosaurid *P. multidentatus* has teeth that, unlike those of *S. gaillardoti*, appear adapted for filter-feeding (*De Miguel Chaves, Ortega & Pérez-García, 2018b*).

In recent years, visualisation with X-ray micro-computed tomography (μCT) has become an increasingly popular instrument for palaeontological investigation, especially to study the internal skull- and braincase anatomy, which are usually not easily accessible (*Miedema*

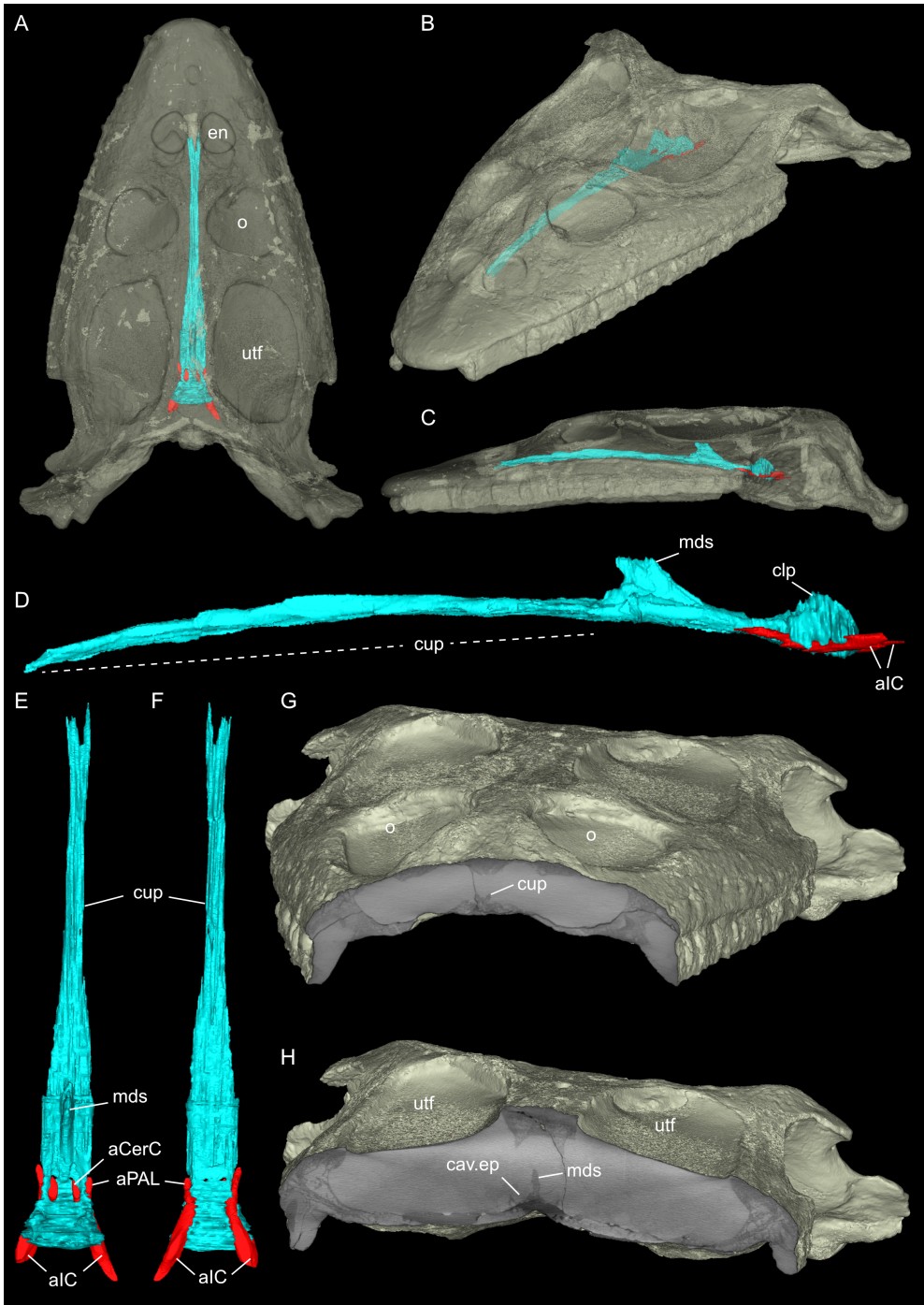

**Figure 2  Virtual rendering and partial segmentation of the full skull of *Simosaurus gaillardoti* (SMNS 16363).** (A–C) Partially transparent skull in showing the position and extent of the parabasisphenoid region, in dorsal (A), angled left anterolateral (B), and left lateral (C) view. (D–F) Segmented parabasisphenoid (in turquoise) and associated major blood vessels (in red), in left lateral (D), dorsal (E), and ventral (F) view. (G, H) Angled surface renderings of skull with virtual sections showing the anterior snout (G) and braincase (H) regions in coronal view. Elements not to scale.

*et al., 2020*; *Sutton, 2008*). Since μCT scanning is a non-destructive technique that reveals internal structures using density differences in the scans (*Germain & Ladevèze, 2021*; *Leyhr, 2023*), this provides improved insight into the morphology and internal anatomy of diverse fossils without damaging or sacrificing rare specimens. Utilisation of μCT scanning thus prevents rare specimens from being destroyed by *e.g.*, acid-preparation (*e.g.*, *Rieppel, 1994b*) or (serial) thin sectioning (*e.g.*, *Klein, 2010*), thus conserving them for improved future methods, while the virtual data can be made easily available and shared without the need to handle the often very delicate and fragile specimens (*Sutton, Rahman & Garwood, 2014*; *Sutton, Rahman & Garwood, 2016*). The number of scanned sauropterygian skulls, including segmented braincases in publications, remain scarce however (summarised for example in *Allemand, Moon & Voeten, 2023*). Sauropterygian skulls that were scanned with μCT include, for example, the nothosaurid *Nothosaurus marchicus* (*Voeten et al., 2018*; *Voeten, Albers & Klein, 2019*), the possible nothosauroid *Hispaniasaurus cranioelongatus* (*Marquez-Aliaga et al., 2017*), the basal eosauropterygian/pachypleurosaur *Keichousaurus hui* (*Liao et al., 2021*; *Xu et al., 2025*), the placodonts *Placodus gigas* (*Neenan & Scheyer, 2012*), *Henodus chelyops* (*Pommery et al., 2021*), *Parahenodus atancensis* (*De Miguel Chaves et al., 2020*), as well as the plesiosaurs *Dolichorhynchops* sp. (*Sato et al., 2011*) and *Libonectes morgani* (*Allemand et al., 2019*). Of the mentioned examples, *H. cranioelongatus* lacks the posterior braincase region, and *H. chelyops* was so far imaged only for dental evaluation and are thus not associated with a segmented braincase (*Marquez-Aliaga et al., 2017*; *Pommery et al., 2021*). In contrast, braincase and/or endocast descriptions based on CT scan data exist for *Keichousaurus hui* (*Xu et al., 2025*), although it was not a focus of that study, as well as for *Dolichorhynchops* sp. (*Sato et al., 2011*), and *Libonectes morgani* (*Allemand et al., 2019*).

Concerning nothosauroids, *Rieppel (1994b)* described the anatomy of acid-prepared skulls of *Simosaurus gaillardoti* and *Nothosaurus* sp. Although the acid preparation, a destructive technique, led to a greater understanding of the exterior skull morphology, it failed to capture the comprehensive internal anatomy of *S. gaillardoti* and often thin and fragile bones within the braincase were lost. More recently, the cranial endocast of *Nothosaurus marchicus* (*Voeten et al., 2018*) was described, but detailed renderings of the braincase bones were not included at the time. The present study aims to describe the braincase anatomy by providing renderings of the well-ossified posterior region of the skull alongside the associated endocranial voids of a *Simosaurus gaillardoti* specimen (Figs. 2A–2H; Fig. S1) that is still largely covered in sediment matrix. Utilising μCT scan data and 3D reconstruction (also of an additional specimen of *S. gaillardoti*; see Fig. S2) are used to explore and interpret the similarities and differences relative to other nothosaurs in their anatomical, morphofunctional, and phylogenetic contexts.

## MATERIALS & METHODS

The material studied consists of the cranium of *Simosaurus gaillardoti* (SMNS 16363), which was found in the Upper Muschelkalk of Murr, Baden-Württemberg, Germany. The cranium is approximately 38 cm in length and is almost complete, but parts have

been reconstructed with plaster for display purposes (Figs. 1D–1F; Fig. S1). In addition, to verify the morphology of some of the braincase of *Simosaurus*, an additional specimen (GPIT/RE/09313) was partially segmented. This specimen was mostly prepared out of the sediment matrix except the braincase region. The *Simosaurus* cranial scan data were then also compared to published scan data on *Nothosaurus marchicus* (TW480000375; *Voeten et al., 2018*); *Placodus gigas* (UMO BT 13; *Neenan & Scheyer, 2012*), and the plesiosaurs *Dolichorhynchops* sp. (ROM 29010; *Sato et al., 2011*) and *Libonectes morgani* (D1-8213; *Allemand et al., 2019*). Due to preservational reasons or difficulties related to clear identification of braincase bones, we refrain from making detailed comparisons with the braincase and cranial endocast of *Parahenodus atancensis* and the available CT scan data of *Keichousaurus hui* and *Hispaniasaurus cranioelongatus*.

## Data acquisition

Both crania were scanned with a Phoenix v|tome|x L240 machine at the Accès Scientifique à la Tomographie à Rayons X facility (AST-RX) in the Muséum national d'Histoire naturelle (MNHN), Paris, France. Micro-X-ray tomography was used to scan the SMNS 16363 cranium twice, once to image the complete cranium (Fig. 2) and once to obtain a higher-resolution scan of the braincase region. The complete cranium was scanned with a voltage of 160 kV and a current of 500 mA. The acquired voxel size is 0.12739793 mm. The higher-resolution scan of the braincase was scanned with a voltage of 165 kV and a current of 500 mA, and the voxel size is 0.06218133 mm. The second specimen, GPIT/RE/09313, was scanned using 150 kV at 500 mA, yielding a voxel size of 0.12354325 mm. Part of the scan data were used in a previous study on the evolution of the sauropterygian labyrinth organ (*Neenan et al., 2017*).

## Data processing

Using MIMICS Innovation Suite 24.0 (Materialise, Leuven, Belgium), the individual bones were identified and traced throughout the μCT image stacks, generating a virtual 3D osteology. In SMNS 16363 the bones that could be identified and traced are the parietal, squamosals, pterygoids (plus associated bones dorsal to the pterygoids), basioccipital, supraoccipital, opisthotics, exoccipitals, prootics and parabasisphenoid (Figs. 3–6). Additionally, anatomical voids were virtually extracted and identified as representing nerves, vasculature and other original soft tissue correlates (Figs. 7–9). These include the cranial endocast, inner ear, sinus system, internal carotid arteries, and cranial nerves. The scan also revealed patches of dark, amorphous material, identified as plaster and artificial infilling, which is not easily identified externally in the specimen (Fig. 1; contrast enhanced in the 3D rendering shown in Fig. S1). For the purposes of this study, the posterior-most portion of the braincase, from the upper temporal region backwards, has been segmented in detail. An overview of the entire cranium in lesser detail was segmented and visualised in VGStudio MAX 2024.1 (Volume Graphics, Heidelberg, Germany). The "Paint and Segment" tool, which utilises machine learning, was used to expedite the segmentation process. With the tool, discrete classes were assigned. Classes denote different types of material present within the scan, as characterised through

different radiodensities (expressed in grey levels) and textures. The chosen classes for this scan were air, bone, matrix, and plaster. For every chosen class, a small part of the skull or otherwise representative material within the scan was segmented. The algorithm subsequently explores the rest of the volume, predicts which sections represent assigned classes, and automatically segments these accordingly. Iterative improvements to correct for misclassification were manually implemented until satisfactory results were achieved. The 'create ROIs' (region of interest) tool was then used to create a 3D model of the cranium with the assigned classes as different regions of interest. The segmenting tools allowed the matrix between the teeth to be removed, after which the model was checked for any major inaccuracies. In addition, the parabasisphenoid and associated passages of the internal carotid arteries were also segmented in the entire skull scans of SMNS 16363 and GPIT/RE/09313; to reconstruct the complete antero-posterior extend of that bone within the crania (Fig. 2; Fig. S2).

## RESULTS

### Systematic palaeontology

Sauropterygia *Owen, 1860*
Eosauropterygia Rieppel, 1994 (*Rieppel, 1994a*)
Nothosauroidea *Nopcsa, 1928* [p. 172; not *Baur, 1889*]
Simosauridae *Huene, 1948*
*Simosaurus gaillardoti Von Meyer, 1842*

### Osteological aspects

As in other sauropterygian reptiles (*e.g.*, *Edinger, 1921*; *Rieppel, 1994b*; *Rieppel, 2000*; *Allemand, Moon & Voeten, 2023*), the braincase of *Simosaurus gaillardoti* is formed by the basioccipital, supraoccipital, a pair of opisthotics, a pair of prootics, a pair of exoccipitals and the parabasisphenoid. These bones are bordered on the dorsal side by the parietal and squamosal forming the posterior skull roof and on the ventral side by the pterygoids (Figs. 3–6). *Rieppel*'s (*1994b*) description of the braincase of *S. gaillardoti* presented the bones, foramina and other structures, which are used as a basis for the current study.

In SMNS 16363, the basioccipital, together with the exoccipitals, forms the ventral and ventrolateral borders of the foramen magnum. The basioccipital can be divided into two parts; the occipital condyle that protrudes straight posteriorly from the cranium on the posterior side and the internal bone that rests on the pterygoid at the posterior border of the cranium Fig. 5). It delimits the cranial endocast ventrally and supports the exoccipitals that project dorsolaterally from the basioccipital. Anteriorly, there is a clear gap separating the basioccipital from the parabasisphenoid, although an *in vivo* cartilaginous connection cannot be ruled out. The exoccipitals extend between the basioccipital and the supraoccipital, form the ventrolateral borders of the foramen magnum, and are laterally flanked by the opisthotics. The exoccipitals participate in the occipital condyle

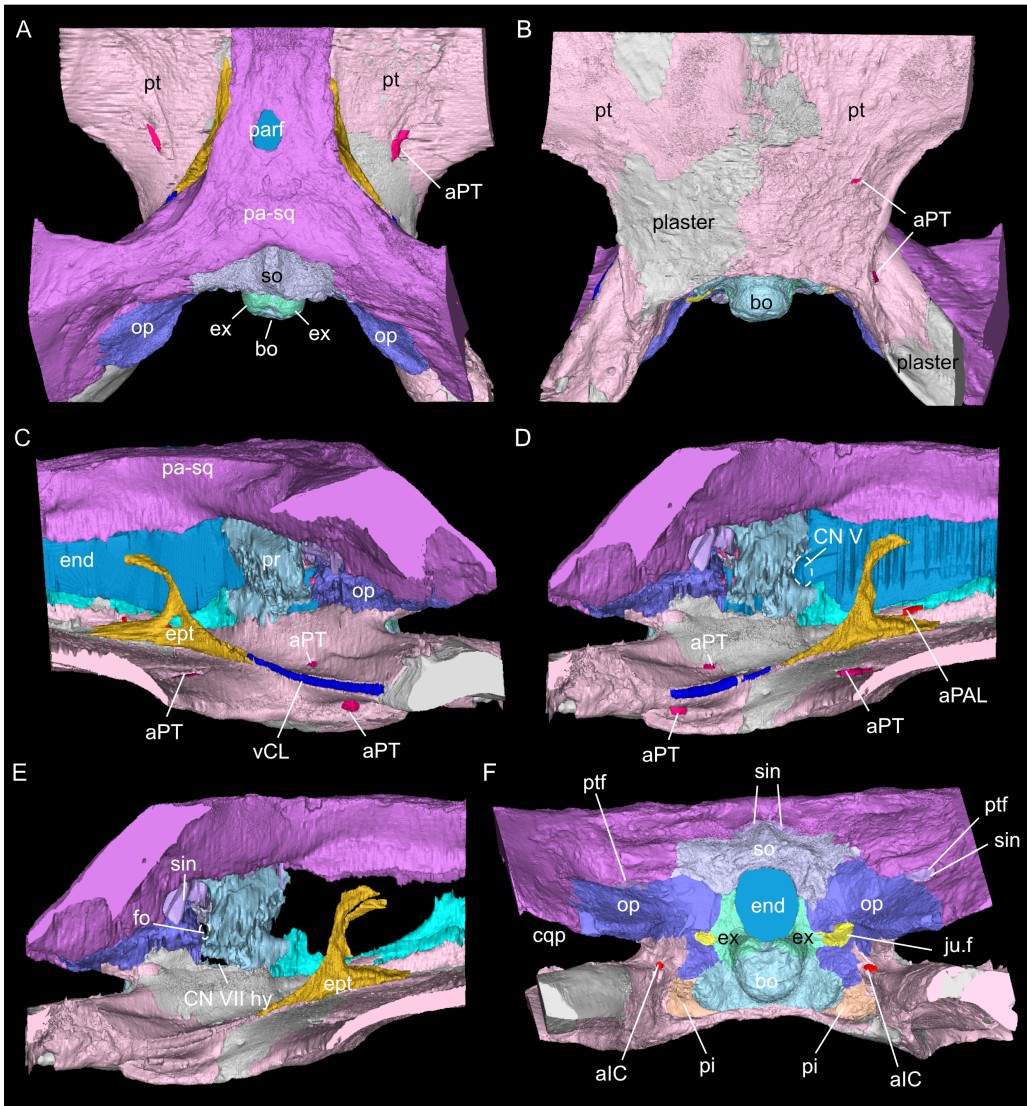

**Figure 3 Virtual rendering and segmentation of the full braincase region of *Simosaurus gaillardoti* (SMNS 16363).** Braincase region in dorsal (A), ventral (B), left lateral (C), right lateral with (D) and without cranial endocast and rendered blood vessels (E), and occipital view (F). Elements not to scale.

dorsolaterally, but they are mostly separated from each other by the basioccipital. The exoccipital bone forms the medial border of one large foramen; the jugular foramen (Fig. 3), also called the metotic foramen (*sensu Rieppel, 1994b*), and two smaller foramina. Laterally, the exoccipitals connect with the opisthotics. The sutures between these bones are difficult to trace on the surface while in the scan data of SMNS 16363, an abutting contact between the exoccipitals and the opisthotics is discernible.

The supraoccipital defines the dorsal border of the foramen magnum and its smooth, concave ventral surface saddles the endocast. It has a medial crest with lateral extensions flanking it on both sides. Anteriorly, this supraoccipital crest bifurcates, creating a Y-shape

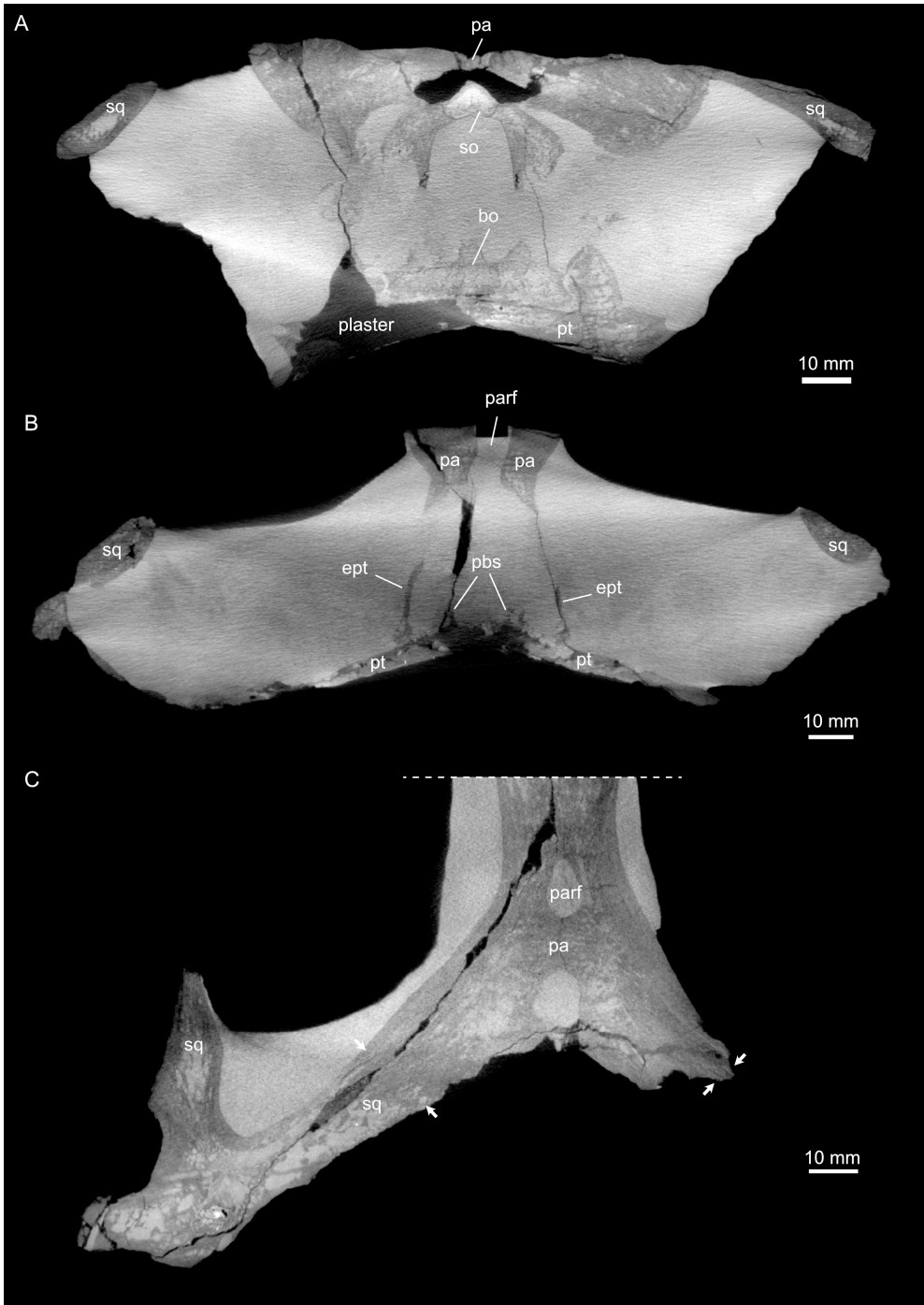

**Figure 4   Micro-CT slice data of *Simosaurus gaillardoti* (SMNS 16363).** (A) Coronal view of braincase region (A) indicating that the bone loss in the specimen occurred mainly due to the breakage of the specimen, as the filled in plaster closely follows the outlines of the preserved bones. (B) Coronal view through braincase at the level of the parietal foramen. The two epipterygoids are clearly visible in articulation with the underlying pterygoids. (C) Axial view of the skull roof bones. Note that the sutures between parietal and the squamosals are only partially visible, potentially indicating incipient fusion of the bones.

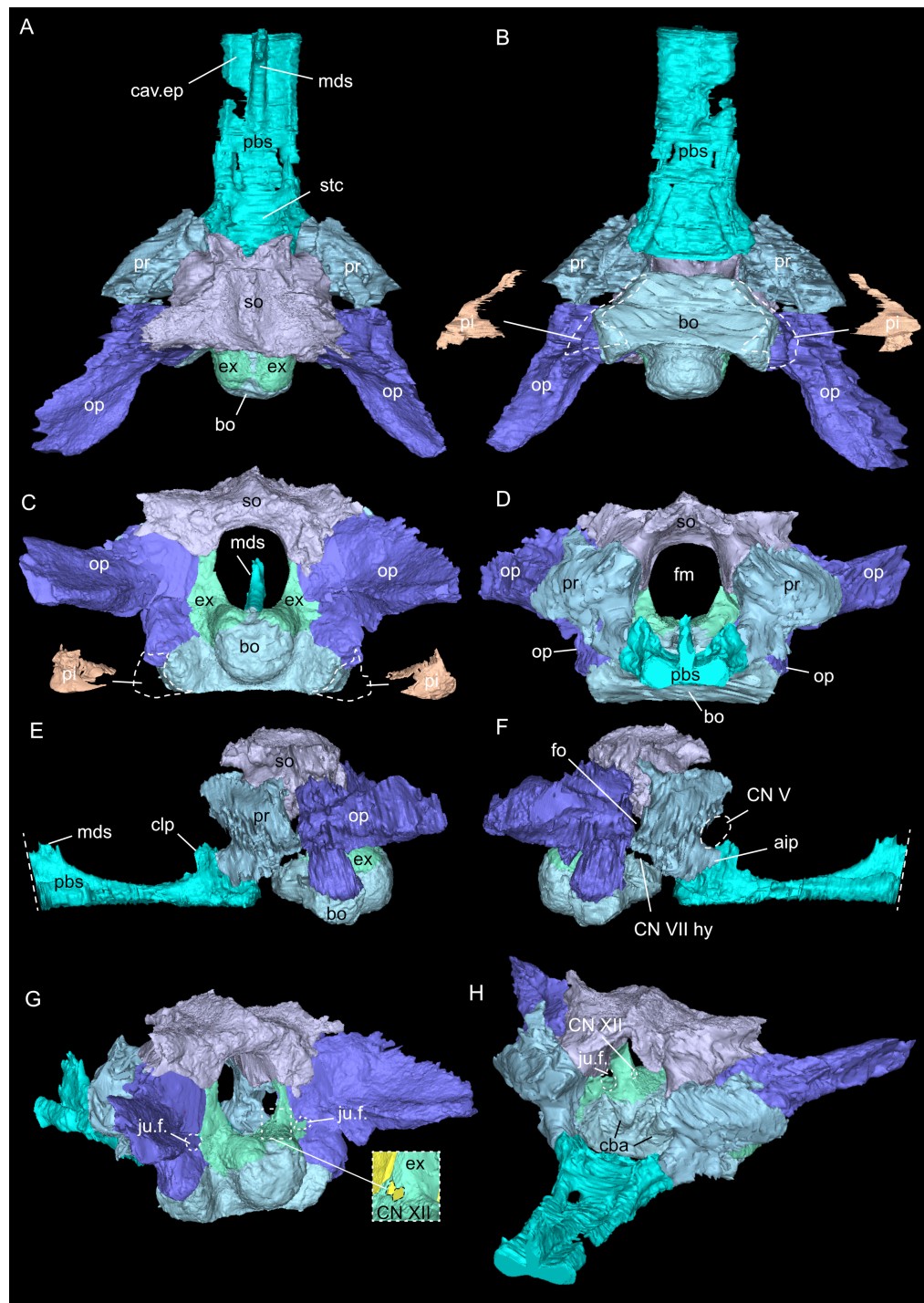

**Figure 5  Virtual rendering and segmentation of bony elements forming the braincase of *Simosaurus gaillardoti* (SMNS 16363).** Braincase in dorsal (A), ventral (B), occipital (C), rostral (D), left lateral (E), right lateral (F), angled right posterolateral (G), and angled left anterolateral view (H). Note that in (B) and (C), the position and the segmented voids of the paracondylar interstices (continued on next page...)

**Figure 5 (…continued)**
are indicated as well (dorsal view in B; occipital view in C). Straight dashed lines in (E) and (F) indicate anterior delimitation of the parabasisphenoid in this higher resolution-, but spatially limited micro-CT scan. The square white stippled frame in (G) indicates where the foramina for cranial nerve CN XII are situated in the right exoccipital (seen in the inset). It is not clear from the scan data whether the foramina are confluent or separated by a thin sheath of bone. Elements not to scale.

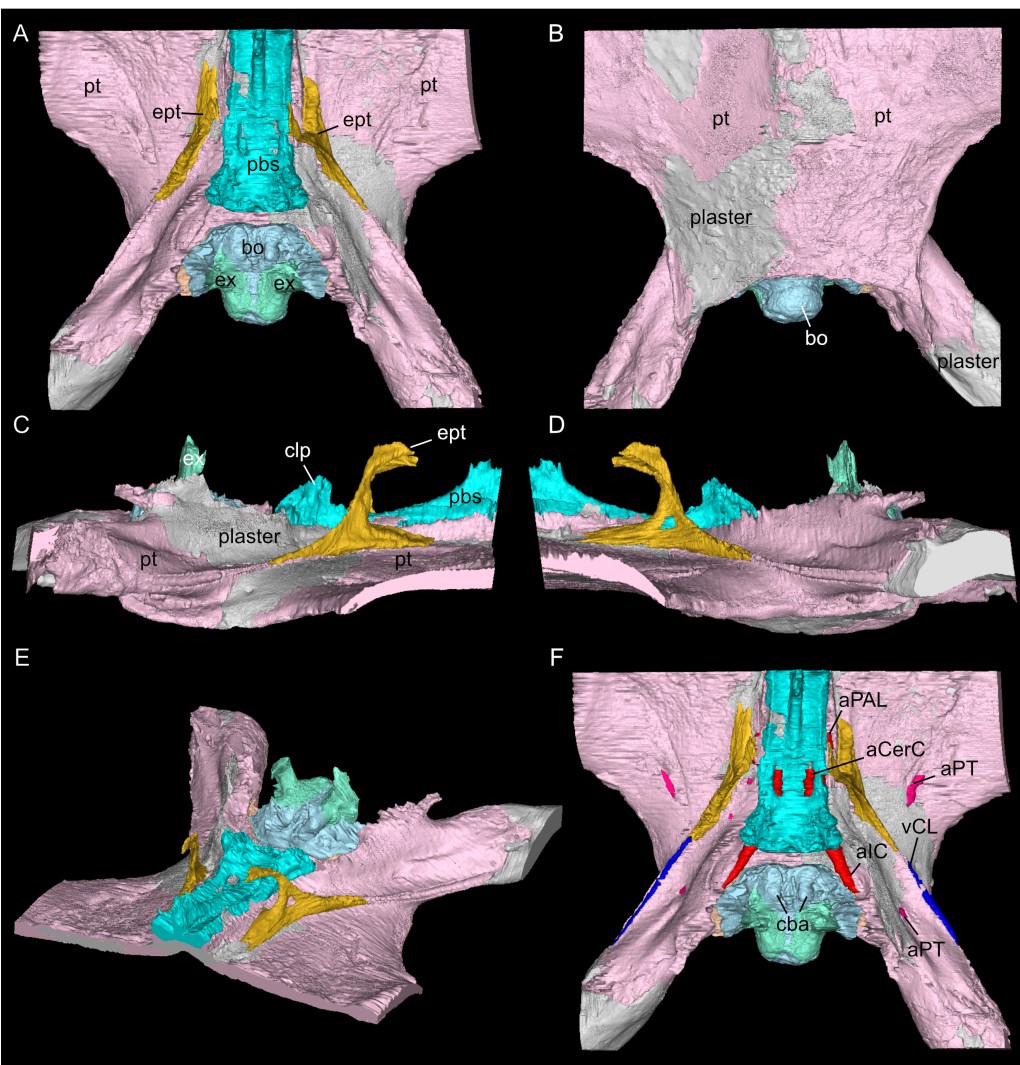

**Figure 6 Virtual rendering and segmentation of the floor of the braincase of *Simosaurus gaillardoti* (SMNS 16363).** Braincase floor elements in dorsal (A), ventral (B), left lateral (C), right lateral (D), angled left anterolateral (E), and dorsal view (F), the latter with segmented blood vessels shown. Elements not to scale.

in dorsal view (Fig. 5A). The lateral surface is very irregular and the left and right sides are not completely symmetrical in shape (Figs. 5C, 5D). The opisthotics connect with the exoccipitals medially, with the supraoccipital dorsomedially, with the basioccipital ventromedially and rest on the pterygoid laterally. On the dorsal and posterolateral side,

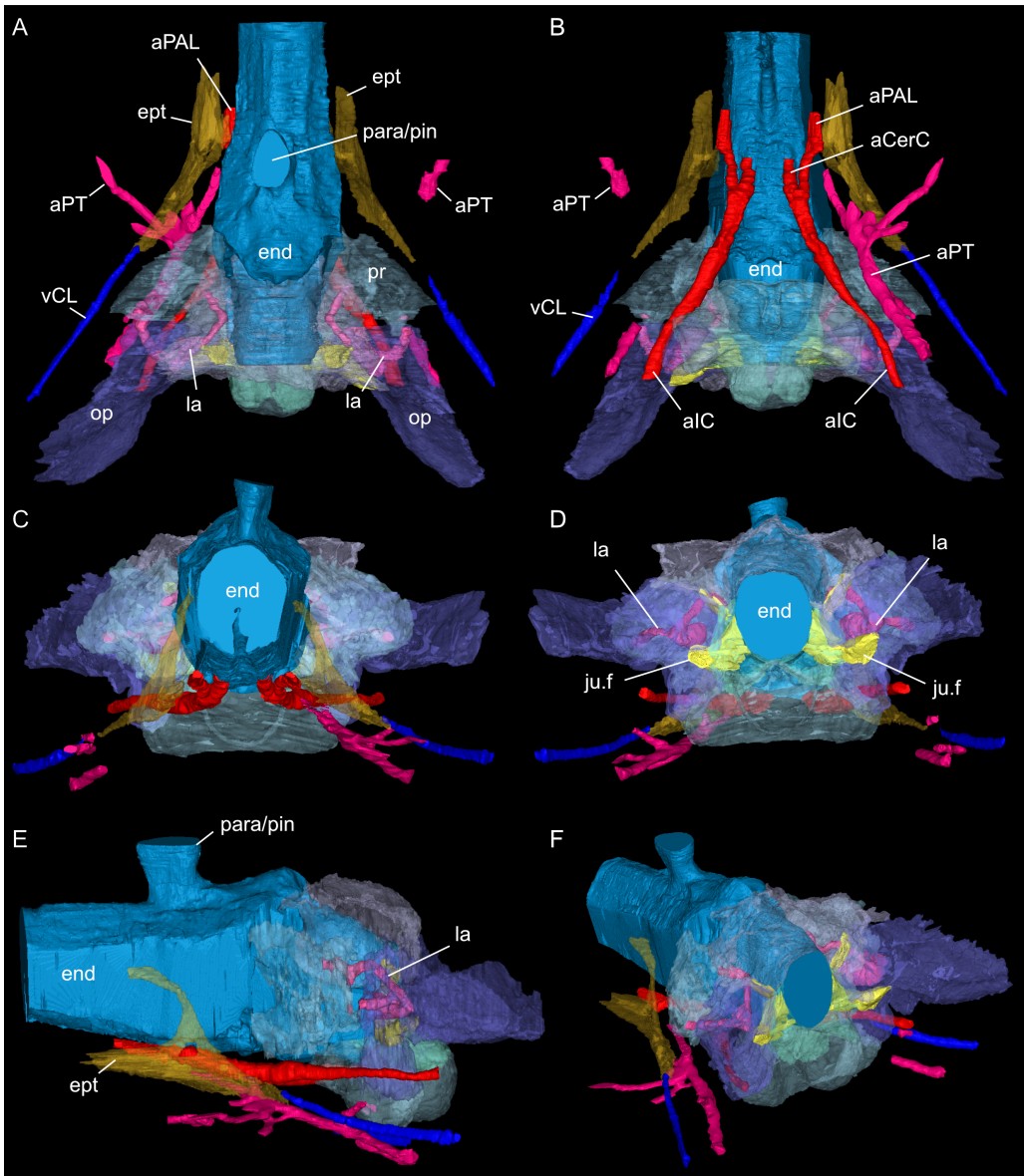

**Figure 7** Segmented posterior cranial endocast and casts of blood vessels, nerves and bony labyrinth of *Simosaurus gaillardoti* (SMNS 16363) surrounded by selected transparent braincase bones. Endocast and selected endocranial voids in dorsal (A), ventral (B), rostral (C), occipital (D), left lateral (E), and angled left posterolateral view (F). Elements not to scale.

the opisthotic meets the squamosal (Fig. 3), and the posttemporal fenestra perforates the suture between these two bones. The opisthotic also forms the dorsal, lateral and ventral margins of the jugular foramen. The opisthotic meets the prootic on a small plane further anterior in the skull, which is visible only in lateral view (Figs. 5E, 5F). The fenestra ovalis (= fenestra vestibuli) is formed between the opisthotic and the prootic. No structure was found in SMNS 16363 that would indicate a stapes articulating with the fenestra ovalis. The prootic meets the supraoccipital dorsolaterally, but it is not visible in posterior view

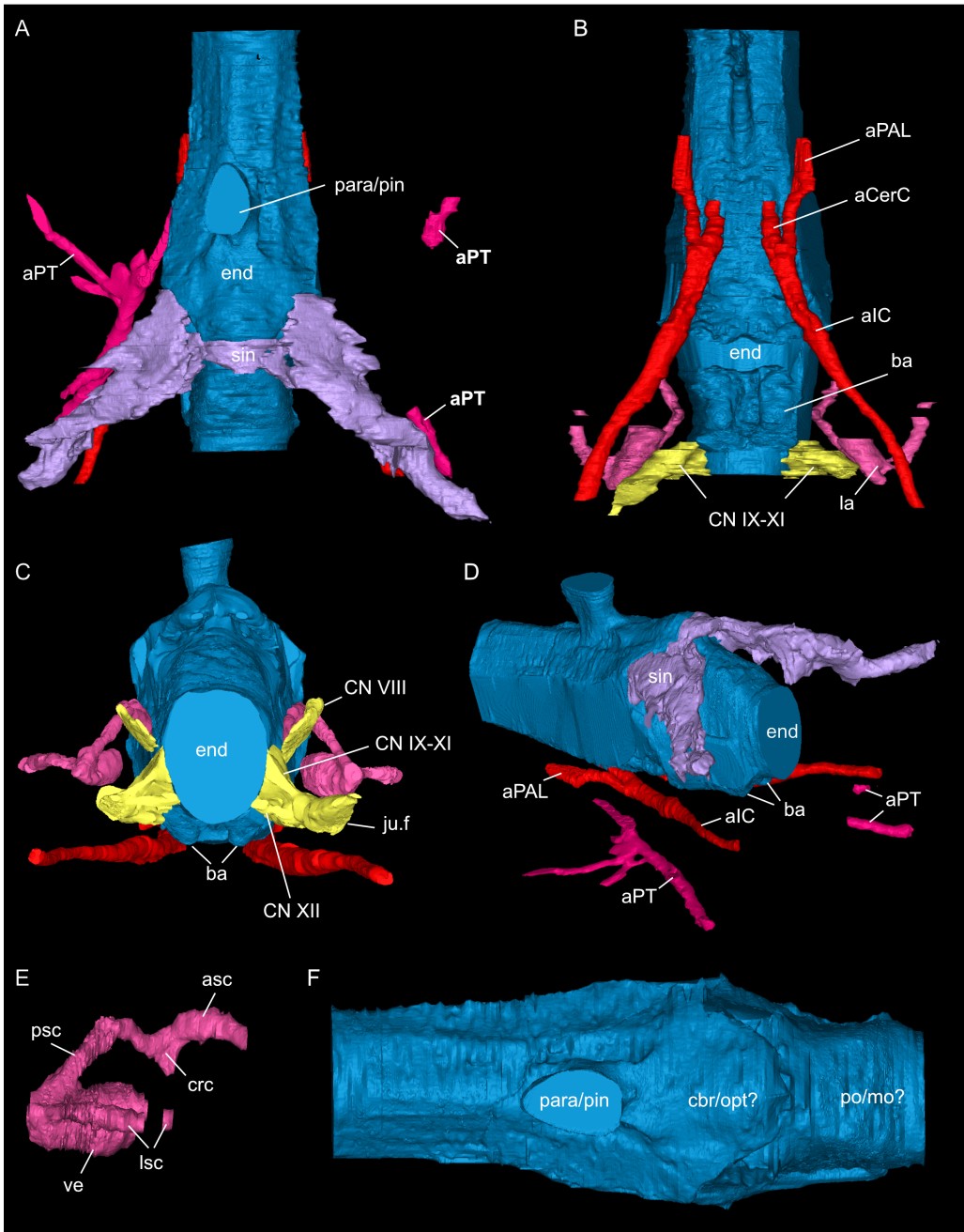

**Figure 8  Segmented posterior cranial endocast and selected cranial voids (blood vessels, nerves, bony labyrinth, and part of sinus system) of *Simosaurus gaillardoti* (SMNS 16363).**  Endocast and selected endocranial voids in dorsal (A), ventral (B), rostral (C), angled left posterolateral (D) views. (E) Right bony labyrinth in medial view. (F) Isolated cranial endocast in dorsal view. Elements not to scale.

as it is obscured by the opisthotic, supraoccipital, and exoccipital in SMNS 16363. The prootic is a round-shaped element with a spongy internal bone structure that houses part of the vestibular system. The prootic meets the parietal dorsally, the supraoccipital

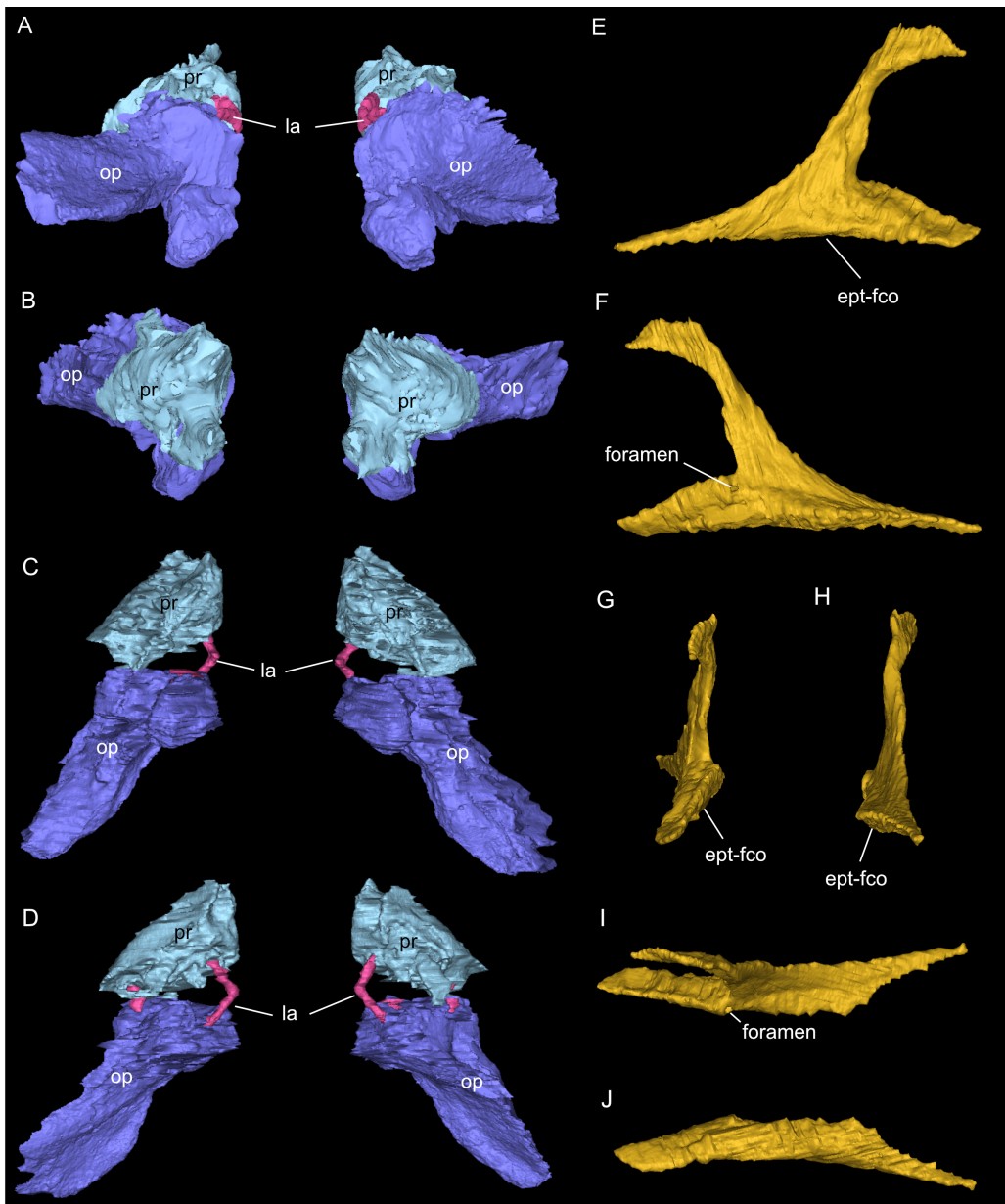

**Figure 9** Segmented associated prootic, opisthotic and bony labyrinth (A–D), as well as right epipterygoid (E–J) of *Simosaurus gaillardoti* (SMNS 16363). Inner ear region in occipital (A), rostral (B), ventral (C), and dorsal view. Right epipterygoid in lateral (E), medial (F), rostral (G), occipital (H), dorsal (I), and ventral (J) view. Elements not to scale.

posterodorsally, the opisthotic posteroventrally and it has an anterior inferior process that meets the parabasisphenoid ventromedially. The anterior border of the prootic frames the oval opening for the trigeminal cranial nerve (CN V) ventrally, posteriorly and dorsally (Fig. 5E; the anterior border of the opening remains free as there is no ossified laterosphenoid in SMNS 16363). The parabasisphenoid consists of two sections:

the posterior section includes the sella turcica, posteriorly bounded by a low dorsum sellae, and the anterior section forms wing-like lateral projections and a narrow medial process that projects anteriorly (Figs. 2, 5 and 6). The sella turcica (Fig. 5A) is a saddle-shaped structure of the basisphenoid on the ventral side of the cranium, framed by bilateral (antero)dorsal extensions that are identified as the parabasisphenoid clinoid processes. As noted already for other *Simosaurus* specimens in *Rieppel (1994a)*; *Rieppel (1994b)*, the sella turcica does not show a median dorsal ridge in the specimens studied herein. Due to damage and subsequent plastering of the pterygoids in SMNS 16363, the anterior extension of the parabasisphenoid and its relationship with the pterygoid is locally difficult to interpret. The wing-like lateral expansions and the medial process, expanding into a median septum anteriorly, form two distinctive anteroposteriorly trending trough-like structures (the cavum epiptericum described for *Nothosaurus* and *Simosaurus* by *Rieppel, 1994b*). Two prominent channels traverse the posterior portion of the parabasisphenoid that house the carotid arteries (see below). In GPIT/RE/09313, the parabasisphenoid and arrangement of the carotid passages was found to have overall the same shape as in the SMNS specimen, although in the former, the median septum and the clinoid processes are slightly more anteriorly expanded and inclined (hook-shaped; Fig. S2). Slightly more anteriorly inclined clinoid processes are also visible in SMNS 10360 (*Rieppel 1994b*: fig. 6). In both our CT scans, the median septum shows a slightly concave posterior margin, a flat and narrow dorsal top, and an excavated anterior margin. A Y-shaped structure of the median septum in cross section as described by *Rieppel* (*1994b*: p. 14) could not be observed as both septa were laterally constricted in our studied specimens. The troughs lateral to the median septum are well marked only until the anterior border of the median septum and then taper anteriorly into a cultriform process which extends anteriorly to the level of the external nares. The cultriform process (Fig. 2; Fig. S2) is laterally crested so the dorsal surface is concave (forming a wide dorsal groove along the process). Grooved and tapering cultriform processes were also described by a wide range of taxa, including for example the neodiapsid *Youngina capensis* (*Gardner, Holliday & O'Keefe, 2010*), the sauropodomorph dinosaur *Massospondylus carinatus* (*Chapelle & Choiniere, 2018*), and recently also in the millerettid *Milleropsis pricei* (*Jenkins et al., 2025*). In those taxa, the cultriform process does not reach the level of the external nares as is the case in *Simosaurus*. Towards its tip, the cultriform process seems to bifurcate slightly, which could be due to taphonomic processes. A similar bifurcation was observed in GPIT/RE/09313, but in this specimen the whole anterior palatal region split medially due to taphonomic distortion of the cranium.

The parietal is an unpaired and medially placed bone (Fig. 3) of the skull roof and it encloses the cranial endocast dorsally and dorsolaterally. The parietal foramen is well-defined, being medially positioned in the posterior skull table (posterior to the widest mediolateral expansion of the upper temporal fenestrae).

The suture between the parietal and the squamosal is partially obliterated externally as well as internally, rendering delimited segmentation of these elements impossible (Fig. 4). The posttemporal fenestra, between the squamosal and the opisthotic, leads to the fenestra ovalis (Fig. 5). The pterygoids form the palate in the braincase region (Fig. 6). Breaks in

the pterygoid are filled with plaster mostly on the ventral side (Fig. 6B) thus obscuring large parts of the medial suture between the right and left pterygoid (Figs. 4 and 6). Thus, the two pterygoids have been segmented as a single unit. The pterygoid is a relatively flat bone in ventral view, forming a sturdy posteriorly extending quadrate ramus along which the quadrate is sutured laterally (note that the quadrate is not separately segmented herein because only a small portion was visible in the scan data of the higher resolution scan of the braincase region of SMNS 16363). In dorsal view, a thin u-shaped groove extends in posterolateral-anteromedial direction over the pterygoid surface, which anteriorly gets covered by a thin hook-shaped bone with a broad and narrow ventral base. Extending along both sides of the endocast and extending anterodorsally, these bones are interpreted as epipterygoids (Figs. 3, 4 and 6), which in contrast to *Nothosaurus* do not contact the parietal dorsally (*Rieppel, 1994b*). Within the pterygoid on the posterior side, the foramen for the internal carotid resides (see below). Between the pterygoid and the basioccipital large excavations exist that taper anteriorly (*i.e.,* the paracondylar interstices; Fig. 3F).

## Cranial endocast

Only the posterior portion of the cranial endocast of SMNS 16363, up to the level of the median septum of the parabasisphenoid was segmented out (Figs. 3, 7 and 8).

Here, the endocranial void is delimited mostly by the dorsal, lateral and ventral braincase elements. Anteriorly the endocast is not delimited laterally by bone. As such, the lateral edges of the endocast (Figs. 7 and 8) are interpolated straight lines between the lowest point on any dorsal bone and the highest point on any ventral bone (*Voeten et al., 2018*). The endocast is wider dorsally than ventrally, granting it an oblong shape in both posterior and anterior view (Fig. 8). Posterodorsally, the endocast portion is delimited by the supraoccipital, posterolaterally by the exoccipitals and opisthotics, posteroventrally by the basioccipital, ventrally by the parabasisphenoid, laterally by the prootics (and by the hook-like epipterygoids), and dorsally/dorsolaterally by the parietal. The endocast smoothly transitions into the sinus system posterodorsally (Figs. 8A, 8D), which renders delimitation of the cerebral endocast and adjacent sinusal structures locally challenging. The part of the endocast that is delimited by the parietal foramen projects vertically, where it expands slightly in diameter, and its dorsal portion is slightly offset to the left in SMNS 16363. The endocast is vaulted and relatively smooth posterior to the parietal foramen (which might have housed a pineal/parapineal gland; see *Smith et al., 2018*) and narrows posteriorly. Anterior to the level of the parietal foramen, the cerebral region of the endocast is present but not well defined laterally. On the posteroventral side, the endocast is delimited by the basioccipital, which defines a median groove with two flanking lobes. Anterior to this, the endocast is artificially smoothed during segmentation due to the absence of any ventral constriction. Here, the ventral boundary was interpolated between the basioccipital and the sella turcica. The sella turcica (Fig. 5A) forms an irregular relief on the endocast. Anterior to the sella turcica, the endocast is constricted by plaster and the pterygoid, which produce a bumpy relief until the ventral groove formed by the median septum of the parabasisphenoid is expressed. The internal carotids enter into the cranial

endocast ventrally, and only a few of the posterior brain nerves could be traced to exit the endocast laterally (see below).

## Nerves and blood vessels

The internal carotid arteries enter the cranium through the internal carotid foramen on the quadrate ramus of the pterygoid (Figs. 7 and 8). The internal carotids then extend along the pterygoid dorsally and ventrally to the parabasisphenoid (Fig. 6F). Distinct ridges are visible on the ventral side of the parabasisphenoid tracing the course of the internal carotid arteries. On the level of the clinoid processes of the parabasisphenoid, the carotid arteries bifurcate (Figs. 7A, 7B). The medial branches are the cerebral carotids, which exist the parabasisphenoid dorsally through two carotid foramina and then merge with the cranial endocast where they cannot be followed anymore. The lateral branches are the palatine arteries. Both palatine arteries can be followed for a short distance more anteriorly, after which they cannot be traced further. The two ventral lobes of the posteroventral endocast (Figs. 8C, 8D) which are expressed as two anteroposteriorly elongate depressions in the basioccipital (Fig. 5H) likely housed paired basilar arteries that enter here into the endocast. Similar paired basilar arteries were discussed for the mosasaur *Platecarpus* (*Russell, 1967*) and have also been segmented out in the braincase of *Plioplatecarpus peckensis* (*Cuthbertson et al., 2015*). The tubular groove (that partly could be roofed over) extending postolaterally-anterormedially over the pterygoid wings is interpreted to potentially have housed the lateral head vein (=vena capitis lateralis; see *Russell, 1967*).

A complex neurovascular system pervades the pterygoid as could be identified on the left side in SMNS 16363 (Figs. 3 and 6–8). On the lateral surface of the quadrate ramus of the pterygoid, a foramen is present (barely visible in ventral view, but well visible in lateral view), identified as the likely entry of an artery into the pterygoid (Figs. 3B–3D), but also potentially housing parts of the palatal branch of the facial cranial nerve (CN VII). This passage within the pterygoid extends as a larger canal that courses anteriorly, parallel to the internal carotid arteries. Smaller branches split off this canal, departing the pterygoid through a number of smaller foramina, three of which open onto the pterygoid dorsally and one foramen on the ventral side. The main canal extends further anteriorly and is visible through the foramen of the CN VII. Slightly posterior to the facial foramen, the canal splits into a complex branching system (leading to the aforementioned smaller foramina), which extends more laterally. At the level of the parietal foramen, the canal exits through the foramen for the presumed palatine branch of CN VII. A similar canal network was likely present also in the right pterygoid (Fig. 7) as evidenced by few small foramina on the dorsal bone surface, but it is only partially preserved due to the breakage of bone and reconstruction with plaster (Figs. 3 and 6).

The jugular foramen is a structure that opens into the endocast and likely accommodated the cranial nerves IX-XII in SMNS 16363 (Figs. 3F, 7). The posteroventral portion likely accommodated CN XII and the more dorsal part for the passage of CN IX-XI (Figs. 8B, 8C). Another structure oblique to the semicircular canals of the inner ear would have connected with the cranial endocast, likely housing the vestibulocochlear nerve (CN VIII).

## Other soft tissue structures

Connected to the posterodorsal portion of the endocast is a large system of cavities (Fig. 8) pertaining to the cranial sinus system (*Witmer et al., 2008*; *Dufeau & Witmer, 2015*; *Voeten et al., 2018*). This system can be followed from the connection to the endocast posteriorly between the skull roof bones (*i.e.,* the parietal and squamosals) and the prootic. It then crosses over into the cavity connected to the posttemporal fenestra between the opisthotic and parietal-squamosal. Within the opisthotics, prootics and supraoccipital, the endosseous labyrinth can only be partially reconstructed in SMNS 16363 (Figs. 7–9). The main body of the labyrinth, the vestibule, is oval-shaped. Most of the posterior semicircular canal (SC) could be reconstructed, meeting the anterior SC in the crus commune, whose ventral connection with the vestibule could not be completely reconstructed (Figs. 8E, 9A–9D). Of the lateral and anterior SC, only the posterior (opisthotic) portions were traceable, whereas the canals could not be traced much further after entering the prootic. As reconstructed, the right and left labyrinth portions in SMNS 16363 generally agree with the morphology seen in the labyrinth extracted from GPIT/RE/09313 (*Neenan et al., 2017*: p. 3853).

## DISCUSSION

### Osteological aspects

Much of *Rieppel*'s (*1994b*) original identification of braincase morphology and interpretations of associated soft tissues can be confirmed by the present study of SMNS 16363 and GPIT/RE/09313. The usage of CT scan data of less or unprepared specimens, however, allowed for the identification and re-interpretation of some cranial structures related to the braincase region.

One of the surprising finds was the identification of clear epipterygoids in SMNS 16363 (Figs. 3, 4 and 6, 9E–9J). In contrast to *Rieppel (1994b)* stating the seeming absence of epipterygoids in *Simosaurus* (based mainly on acid-prepared specimens such as SMNS 10360, 50714 and 50715), two hook-shaped epipterygoids could be identified, whereas their absence in other specimens seems to be thus a preparatory (or preservational) artefact. The left epipterygoid bone has a small perforation in its triangular ventral portion (*i.e.,* the pterygoid facet of the bone; Figs. 3C, 6D, 6E). However, this hole represents an artefact associated with a mechanical fracture (see Fig. 4B) rather than an original anatomical feature. In GPIT/RE/09313, the area where the epipterygoids would have been placed, is clearly damaged (Fig. S2), as is the region in SMNS 10360 and SMNS 16767. The epipterygoids in SMNS 16363 lack a parietal contact and flank the anterior extent of the posterior ossified part of the parabasisphenoid complex. The latter expands into a median septum as indicated by *Rieppel (1994b)*. The anatomical location of the median septum agrees with the ventral-dorsal extension of the parabasisphenoid in SMNS 16363 anterior to the level of the parietal foramen, but a dorsal expansion (into a Y-shaped structure) was not observed in both CT-scanned specimens studied herein. Towards its anterior, the parabasisphenoid extends as a tapering thin and grooved cultriform process up to the level of the external/internal nares. The process thus likely supported the infraorbital to rostral endocast (*i.e.,* the olfactory tract) ventrally, as was also shown for *Nothosaurus marchicus* (*Voeten et al., 2018*).

The excavations lateral to the basioccipital have been referred to as eustachian foramen, as they were interpreted as housing the eustachian tube or the internal carotid arteries in *Nothosaurus* (*Koken, 1893*). As such, the eustachian tube was interpreted as connecting the middle ear and the pharyngeal region in these animals (*Rieppel, 1994b*). As shown by the previous study on *Nothosaurus marchicus* (*Voeten et al., 2018*), these excavations, termed paracondylar interstices in that study, neither housed the internal carotids, nor connected to the pharyngeal region (and thus not serving as a connection with the middle ear). In SMNS 16363, the paracondylar interstices (Fig. 3F) can be followed until the anterior end of the basioccipital, from where they do not connect to the endocast or the middle ear. As in *N. marchicus* (*Voeten et al., 2018*), any tiny spaces extending anteriorly to the paracondylar interstices are due to blunt abutting rather than a tight suturing of the braincase bones in this region.

In SMNS 16363, the medial pillars of the exoccipitals articulate more posteriorly and posterolaterally with the basioccipital, so that we here consider them to participate in the formation of the occipital condyle, in contrast to what has been reported for some other specimens of *Simosaurus gaillardoti* (*Rieppel, 1994a*; *Rieppel, 1994b*).

As discussed by *Rieppel (1994b)*, the location of the fenestra ovalis and the structure of adjacent cranial elements are different between *Simosaurus* (SMNS 10360, 15860, 50714 and 50715) and *Nothosaurus* (SMNS 59075 and 59076). In *Simosaurus*, the fenestra ovalis is located between the opisthotic and prootic and is visible in lateral view, whereas the fenestra ovalis is not visible in lateral view in *Nothosaurus* (*Rieppel, 1994b*). Similar to *Rieppel (1994b)*, no stapes were encountered in the studied specimens of *Simosaurus* herein.

The posttemporal fenestrae are arranged differently in *S. gaillardoti* compared to the reduced posttemporal foramen in *N. marchicus*, which in the latter are medial to the paracondylar interstices (*Voeten et al., 2018*) but are laterally placed in SMNS 16363. *Voeten et al. (2018)* rejected the term 'eustachian foramen' (*sensu Rieppel, 1994b*; *De Miguel Chaves, Ortega & Pérez-García, 2018a*), rendering the placement and geometry of the paracondylar interstices potentially more flexible, while arguing against the functions otherwise associated with an eustachian foramen.

The posttemporal fenestra (Fig. 3F) is located between the opisthotic and squamosal in *Simosauru* s, whereas the reduced posttemporal foramen in *Nothosaurus* is bordered by the exoccipital, supraoccipital and in some cases the opisthotic (*e.g.*, *Rieppel, 1994b*; *Voeten et al., 2018*; *Shang, Li & Wang, 2022*). In SMNS 16363, the posttemporal fenestrae are not as large as observed in *Placodus gigas* (*Sues, 1987*), but they are clearly visible and well-defined. Placodonts were obligately durophagous, feeding primarily on hard-shelled prey (*Rieppel, 2002*). *S. gaillardoti* was also durophagous but likely supplemented its diet by catching fish using rapid snapping motions (*Rieppel, 1994a*). *N. marchicus*, in contrast, captured fish through a quick side-swipe motion with its distinct fish-trap dentition (*Rieppel, 2000*). This morphofunctional gradient coincides with a shift from large and well-defined posttemporal fenestrae in basal Triassic Sauropterygia to reduced or absent posttemporal foramina in nothosauroids. Progressive reduction of posttemporal fenestrae/foramina thus appears complementary to the development or enlarged upper temporal fenestrae

in nothosauroids (*Rieppel, 1994b*) relative to placodonts, which are the sister taxon to Eosauropterygia. Continued specialisation and the establishment of obligate piscivory with clear fish-trap dentition in nothosaurids may explain the difference between the size of posttemporal fenestrae/foramina in *Simosaurus* and *Nothosaurus*, respectively.

## Cranial endocast

The general shape of the endocast of *S. gaillardoti* is similar to that of *Nothosaurus* (Fig. 10), being elongated and tapering anteriorly with an anteroposterior orientation (*e.g.*, *Edinger, 1921*; *Voeten et al., 2018*). The endocast of *P. gigas* is also elongated (Figs. 10C, 10D), but the shape is anteroposteriorly sigmoidal (*Neenan & Scheyer, 2012*). The most striking difference between the endocasts of *S. gaillardoti* (Figs. 10A, 10B) and *N. marchicus* (Figs. 10E, 10F) is the dorsoventral flatness of the latter (*Voeten et al., 2018*). The endocast of *S. gaillardoti* is narrower with a vertically oriented oval shape in coronal view, whereas the endocast of *N. marchicus* has a more horizontally oriented oval shape. The flatness of the *N. marchicus* endocast can be attributed to its dorsoventrally flattened skull inferred to be hydrodynamically specialised for lurching ambush predation (*Voeten et al., 2018*). The less flattened endocast of SMNS 16363 is thus likely associated with its less dorsoventrally flattened skull (which also might reflect a requirement for more robusticity/rigidity in a cranium that engages in facultative durophagy), indicating a more broad-spectrum feeding strategy that included both durophagy and piscivory (*Rieppel, 1994a*). The endocast of *S. gaillardoti* was also compared with the endocast of *Libonectes morgani* (*Allemand et al., 2019*). *L. morgani* is an elasmosaurid with an anteroposteriorly elongated endocast. Unlike *S. gaillardoti* and *N. marchicus*, the endocast of *L. morgani* (Figs. 10G, 10H) is dorsoventrally thick and distinctly thins at the anteroposterior level of the para-/pineal complex. Similarly, the endocast of *Dolichorhynchops* sp. would be posteriorly thickened based on the reconstructed braincase by *Sato et al. (2011)*, although its anterior endocast portion remains elusive due to the lack of preserved bone in ROM 29010. The endocast of *L. morgani* also shows two well-defined lobes on the dorsal side posterior to the parapineal/pineal complex (Figs. 10G, 10H). The posterior lobe was interpreted to represent the cerebellum, and the anterior lobe was associated with the optic lobe (*Allemand et al., 2019*; *Allemand, Moon & Voeten, 2023*). The endocast of *S. gaillardoti* shows a singular lobe posterior to the pineal complex (Fig. 10A), partly segmented as the sinus system. Extrapolating from the condition of *L. morgani*, this lobe could be either the cerebellum or the optic lobe. The optic lobes are placed more laterally in *N. marchicus*, which would suggest the referred lobe is the cerebellum in SMNS 16363. However, the cerebellum is often unpronounced in marine reptiles, and in *Simosaurus* even less developed than in plesiosaurs (*Allemand et al., 2019*). Thus, whether this lobe is the optic lobe, or the cerebellum remains unclear. The boundary between the pons (posterior-most midbrain) and the medulla oblongata (anterior-most hindbrain) could be recognised in *N. marchicus* based on a faint flexure (Fig. 10F; *Voeten et al., 2018*). In *S. gaillardoti*, the endocast has a relatively consistent shape at the posterior-most portion, then expands laterally at the anterior end of the basioccipital, and this could be the border of the medulla oblongata (Fig. 10A). It must be noted that the ventral border between the basioccipital and the

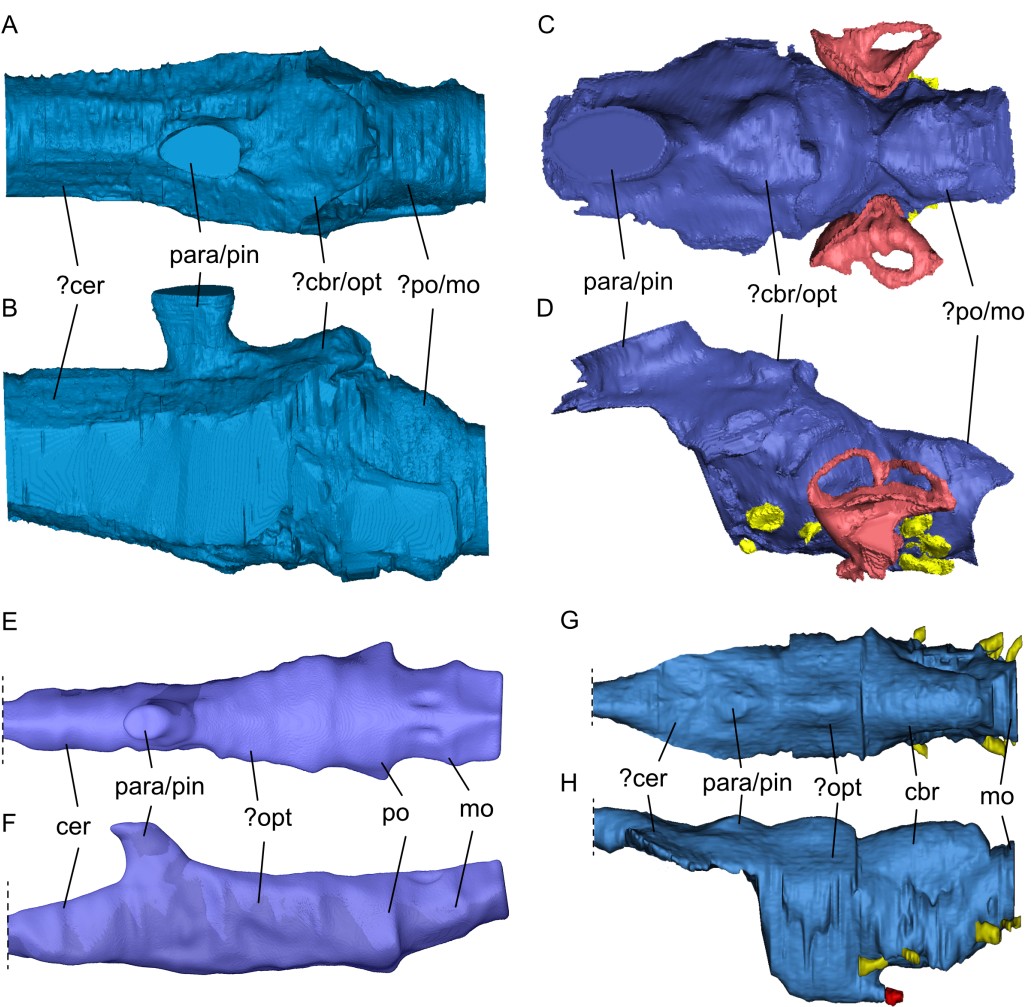

**Figure 10  Comparison of segmented sauropterygian endocasts.** (A, B) *Simosaurus gaillardoti* SMNS 16,363; this study; (C, D) *Placodus gigas* (UMO BT 13; modified after *Neenan & Scheyer, 2012*); (E, F) *Nothosaurus marchicus* (TW480000375; modified after *Voeten et al., 2018*); (G, H) *Libonectes morgani* (D1-8213, modified after *Allemand et al., 2019*; image credit: Rémi Allemand). Note that the cranial endocast of *P. gigas* was mirrored to reflect the direction of the other endocasts (anterior is towards the left). Images in (A), (C), (E), and (G) show the endocasts in dorsal, images in (B), (D), (F), and (H) in lateral view. Passages for cranial nerves are indicated in yellow, blood vessels in dark red, and the bony labyrinth in lighter red/pink. Elements not to scale.

parabasisphenoid is not well delimited due to a gap between both bones (which may have featured a cartilaginous connection during life; Figs. 6 and 7). The boundary between pons and medulla oblongata border could thus also be placed more anteriorly in this transitional region.

## Nerves and blood vessels

The internal carotid artery enters the skull through a small foramen ventrolateral to the jugular foramen (carotid canal; *Rollot, Evers & Joyce, 2021*), which is more dorsally and medially placed in *S. gaillardoti* than in *N. marchicus*. The internal carotid arteries

also split more anteriorly in *S. gaillardoti* than in *N. marchicus* and in plesiosaurs. In the latter, they split even more anteriorly than in *S. gaillardoti*, specifically anterior to the basisphenoid (*Allemand, Moon & Voeten, 2023*). The palatine nerve may have been housed by the canal containing the palatine artery in *S. gaillardoti*, which is the lateral branch after the bifurcation (*Rieppel, 1994b*).

Similar to the condition in *Nothosaurus* spp. (*Rieppel, 1994b*; *Voeten et al., 2018*) only the para- and basisphenoid complex region is ossified and fused in *Simosaurus gaillardoti*, whereas the more lateral/dorsal sphenoid regions remain un-ossified (*i.e.,* a laterosphenoid is absent). As such, the cranial nerves I-IV can also not be reconstructed using the CT scanned specimens of *Simosaurus* and only a portion of the CN V is framed by the prootic anterior inferior process ventrally. Although *Neenan & Scheyer (2012)* also indicated the absence of a laterosphenoid in *Placodus gigas*, a complete prootic fenestra is shown, surrounded by the bone of the prootic (including the anterior inferior process). Alternatively, the anterior and anterodorsal part of the fenestra might not be constructed by the prootic but by a thin laterosphenoid instead. A similar condition was, for example, also described in the early branching archosauromorph *Tanystropheus hydroides* (*Spiekman et al., 2020*). The laterosphenoid is generally considered a typical structure found in taxa of the archosaur lineage including extant crocodylians and birds (*e.g.*, *Kuzmin et al., 2021*), although *Rieppel (1976)* argued for referral of the bone in archosaurs as pleurosphenoid instead. In the latter study, the laterosphenoid of snakes was considered to be a non-homologous structure to that of archosaurs, based on differences in cranial development. The term laterosphenoid seemed to have gained traction in archosaur research, however, and a morphologically similar bone to that in the archosaur cranium has been proposed as a homologous element in the cranium of the stem turtle *Proganochelys quenstedtii* as well (*Bhullar & Bever, 2009*). *Evers, Barrett & Benson (2019)*, however, argued that the fenestra prootica could not be considered a homologous structure to the trigeminal foramen, a secondarily formed structure in 'anatomically modern turtles' (*Scheyer et al., 2022*), further raising questions of homology of this part of the reptilian braincase feature, but a review of this feature would go beyond the scope of this article.

*Rieppel (1994b)* hypothesised that the palatine branch of the facial nerve could have emerged from the facial foramen and entered the foramen for the palatine branch of the facial nerve (VIIpl). The course of the facial nerve is unclear in our observed specimen, which does not allow for testing the proposed course of the palatine branch.

According to *Rieppel (1994b)*, each exoccipital in *S. gaillardoti* would carry three foramina, a very large one crossing the exoccipital in anteromedial-posterolateral fashion, and two smaller foramina that pierce the pillar-like structure of the exoccipital from medial to lateral. The largest of the three foramina was identified as the jugular foramen by *Rieppel (1994b)*, also referred to as the metotic foramen in that work, whereas the two smaller foramina, of which the anterior one was found to be a bit larger and the posterior one a bit smaller, open up laterally into the jugular foramen and were interpreted to have housed two branches of the hypoglossal nerve (CN XII). The scan data of SMNS 16363 studied here in detail confirms the hypoglossal foramina that open medially into the braincase just anterior to the posterior border of the foramen magnum and connect

laterally with the large jugular foramen. The specimen seems to show a slight asymmetry, however, in having only a single ovoid shaped hypoglossal foramen in the left exoccipital and a horizontal 'figure-eight' shaped foramen in the right exoccipital. These expressions would agree with either a single confluent foramen or two foramina that are separated by a very thin bony sheath. Such a distinction cannot be confidently inferred from the CT scan data, nor could a relative size difference between the presumed larger anterior and smaller posterior hypoglossal foramen be conclusively confirmed. *Rieppel (1994b)* considered the large jugular foramen completely enclosed by the exoccipital, which, in a composite interpretative drawing of several *Simosaurus* specimens, was interpreted to have a square shape with an almost vertical, broad sutural contact with the opisthotics. This bone arrangement precluded a contact between the opisthotics and the basioccipital in posterior view. The scan data of SMNS 16363 indicate, however, that the exoccipitals did not have a broad lateral projection and that the jugular foramina were framed as well by the opisthotics, the latter reaching the basioccipital by a broad descending projection ventromedially (Figs. 3 and 5). A broader connection between occipital and opisthotics was also shown in *Nothosaurus cristatus* (*Hinz, Matzke & Pfretzschner, 2019*).

According to *Rieppel (1994b)*, the foramen piercing the prootic ventral to the fenestra ovalis could have housed the hyomandibular branch of the facial nerve (VIIhy). The prootic in SMNS 16363 is porous and features a large foramen that correlates with *Rieppel*'s (*1994b*) observations.

Of the cranial nerves exiting the posterior portion of the brain, the glossopharyngeal (CN IX) and vagus (CN X) cranial nerves are intimately related and have been interpreted to enter through the jugular foramen in *N. marchicus* (*Voeten et al., 2018*). It seems possible that these nerves, together with the hypoglossal (CN XII) nerve (and the spinal accessory nerve CN XI) passed through the jugular foramen in *S. gaillardoti*. This condition would differ from *P. gigas*, in which these nerves do not enter through the same foramen (*Neenan & Scheyer, 2012*). The hypoglossal (CN XII) nerve in *P. gigas* exits through a foramen in between the exoccipital and the basioccipital (potentially a derived condition in placodonts), and the glossopharyngeal (CN IX) and vagus (CN X) nerves pass through the jugular foramen. The location where the nerves enter the cranial endocast is different between *S. gaillardoti* and *N. marchicus*. The nerves exit more ventrally in *N. marchicus* than in *S. gaillardoti*. In *S. gaillardoti*, the nerves merge into the endocast just anterior to the posterior border of the foramen magnum. The course of the nerves can also be followed further anterior in *N. marchicus* (*Voeten et al., 2018*), where they merge with the endocast ventral to the foramen magnum.

## Other soft tissue structures

The large dorsal system connected to the endocast is likely the sinus system or houses parts of the middle cerebral vein (*Voeten et al., 2018*). The structure has a similar placement in *N. marchicus* but has a smoother transition into the endocast in *S. gaillardoti* than in *N. marchicus*, where it emerges relatively perpendicularly to the endocast.

The inner ear has been shown to have adaptations to aquatic life, where the semicircular canals have decreased height and/or increased width (*Georgi & Sipla, 2008*). *Neenan et al.*

*(2017)* showed that the inner ear altered with changes in locomotion in marine reptiles. The inner ear of *Placodus gigas* was distinctly larger than the inner ears of all other members of Sauropterygia, following also general shape changes with the transition from near-shore swimmers such as *Nothosaurus* spp. and *S. gaillardoti* to pelagic swimmers such as *Callawayasaurus colombiensis* and *Libonectes morgani*. The endosseous labyrinth of plesiosaurs was found distinctly more compact, showing anteroposterior shortening and thickening of the semicircular canals (*Neenan et al., 2017*). These adaptations align with the morphology of the endosseous labyrinths of an extant crocodile (*Crocodylus acutus*) and an extant sea turtle (*Lepidochelys olivacea*). According to *Neenan et al.* (*2017*: p. 3855), the inner ears of the extinct near-shore swimmers resembled that of the crocodylian in having more "long, narrow semicircular canals, a narrow crus communis, a taller anterior canal, and a general rounded or pyramidal shape in lateral view that creates an M-shaped outline", whereas the pelagic swimmers look more similar to that of the extant sea turtle "in having shorter semicircular canals with wider cross-sectional diameters, anterior and posterior canals that are roughly equal in height, and a wide crus communis". The inner ear of SMNS 16363 (Fig. 8E) has a similar morphology to that of GPIT/RE/09313, the *Nothosaurus* sp. specimen (NME 16/4), and the *Augustasaurus hagdorni* pecimen (FMNH PR 1974) used in *Neenan et al. (2017)* and thus reflects the adaptations for aquatic life in near-shore environments.

## Palaeoecological implications

SMNS 16363 has a braincase morphology that is quite similar to that of *Nothosaurus marchicus* (specimen TW480000375). The two share similar locations of fenestra, such as the posttemporal fenestra, and overall attachments of bones, such as the placement of the exoccipitals, supraoccipital and opisthotics (*Rieppel, 1994b*). These conditions are markedly different in placodonts. Additionally, internal structures such as the (para-)pineal complex and the endocast have a broadly similar shape (Fig. 10). The morphology of the inner ear in *Nothosaurus* sp. and *Simosaurus* shares adaptations for near-shore aquatic life as they both feature a smaller endosseous labyrinth than placodonts, but are dorsoventrally more elongated semicircular canals compared to the pelagic plesiosaurs (*Neenan et al., 2017*). Based on these similarities, the present study supports a closer relationship of *Simosaurus* with other nothosaurs within Nothosauroidea. The near-vertical projection of the parapineal/pineal complex in *Simosaurus* relative to its subtly forward-slanting orientation in *Nothosaurus* (*Voeten et al., 2018*) could reflect a slightly different mode of cranial development to facilitate the more specialised derived cranial architecture in the latter. Nothosauria have been interpreted as piscivorous ambush predators with fish-trap dentition (*Rieppel, 2002*; *Voeten et al., 2018*), whereas *Simosaurus gaillardoti* has a more generic (opportunistic) feeding strategy, including facultative durophagy (*Rieppel, 2002*). This is supported here by the dorsoventrally flattened endocast of *Nothosaurus marchicus* (TW480000375; *Voeten et al., 2018*) relative to the less dorsoventrally flattened endocast (and skull) of SMNS 16363. Durophagy is also exhibited by the near-shore Placodontiformes which is usually regarded as the most basal sauropterygian clade (*Rieppel, 1999*; *Rieppel, 2000*; *Rieppel, 2002*). Like nothosaurs, pistosaurs have been interpreted as

piscivorous (*Storrs, 1993*). Thus, depending on which phylogenetic scenario is used (see Fig. 1), piscivory with clear fish-trap dentition can be interpreted to have developed multiple times independently in several sauropterygian clades, but for now the discussion remains open due to lack of a conclusive systematic framework of Sauropterygia.

## CONCLUSIONS

The braincase of *S. gaillardoti* had only been described exteriorly, using acid-prepared skulls and only a limited number of 3D segmented braincases of other Sauropterygia exist. The virtually segmented braincase of SMNS 16363 (and that of the partly segmented GPIT/RE/09313) gives new valuable information not available previously for *Simosaurus gaillardoti*. This provides a better understanding of the internal morphology of the bones and soft tissue structures, such as cranial nerves and vasculature. It also indicates the existence of epipterygoids in *S. gaillardoti* not observed earlier in acid-prepared specimens. The inner ear of SMNS 16363 is comparable to that of other near-shore sauropterygians in overall shape, sharing adaptations for aquatic life but in shallower aquatic settings than the open-water plesiosaurs as previously indicated (*Neenan et al., 2017*). The braincase anatomy and brain endocast support a phylogenetic placement of *Simosaurus gaillardoti* close to other nothosauroids. Also, piscivory is the currently proposed feeding strategy for most nothosaurs and pistosaurs but not for *S. gaillardoti* and *Paludidraco multidentatus*. Thus, clear fish-trap dentition piscivory could be potentially independently derived in nothosaurs and pistosaurs. Future research on more related taxa, along with additional scans and segmented endocasts across the phylogeny will provide additional, valuable information about sauropterygian phylogenetic relationships and their evolutionary history.

### Institutional abbreviations

| | |
|---|---|
| **D1** | collections of the Rhinopolis Museum, Gannat, France |
| **GPIT** | Paläontologische Sammlung der Universität Tübingen, Germany |
| **ROM** | Royal Ontario Museum, Toronto, Ontario, Canada |
| **SMNS** | Staatliches Museum für Naturkunde Stuttgart, Germany |
| **TW** | collections of Museum TwentseWelle, Enschede, The Netherlands |
| **UMO** | Urwelt-Museum Oberfranken, Bayreuth, Germany |

### Anatomical abbreviations

| | |
|---|---|
| **aCerC** | cerebral carotid artery |
| **aIC** | internal carotid artery |
| **aip** | anterior inferior process of prootic |
| **aPAL** | palatine artery |
| **aPT** | pterygoid artery |
| **asc** | anterior semicircular canal |
| **ba** | basilar artery |
| **bo** | basioccipital |
| **cav.ep** | cavum epiptericum |

| | |
|---|---|
| **cba** | cavity for basilar artery |
| **cbr** | cerebellum |
| **cer** | cerebrum |
| **clp** | clinoid process of parabasisphenoid complex |
| **CNV** | passage for trigeminal cranial nerve (V) |
| **CN VII hy** | passage for hyomandibular branch of facial cranial nerve (CN VII) |
| **CN VIII** | passage for vestibulocochlear cranial nerve (VIII) |
| **CN IX-XI** | passage for glossopharyngeal (IX), vagus (X) and accessory cranial nerves (XI) |
| **CN XII** | passage for hypoglossal cranial nerve (XII) |
| **crc** | crus commune |
| **cup** | cultriform process |
| **cqp** | cranio-quadrate passage |
| **en** | external narial opening |
| **end** | endocast |
| **ept** | epipterygoid |
| **ept-fco** | pterygoid facet of the epipterygoid |
| **ex** | exoccipital |
| **fm** | foramen magnum |
| **fo** | fenestra ovalis |
| **ju.f** | jugular foramen |
| **la** | bony labyrinth |
| **lsc** | lateral semicircular canal |
| **m** | sediment matrix |
| **mds** | median septum |
| **mo** | medulla oblongata |
| **o** | orbit |
| **op** | opisthotic |
| **opt** | optic lobe |
| **pa** | parietal |
| **para/pin** | parapineal/pineal complex |
| **parf** | parietal foramen |
| **pa-sq** | undifferentiated parietal-squamosal |
| **pbs** | parabasisphenoid |
| **pi** | paracondylar interstices |
| **po** | pons |
| **pr** | prootic |
| **psc** | posterior semicircular canal |
| **pt** | pterygoid |
| **ptf** | posttemporal fenestra |
| **sin** | sinus system |
| **so** | supraoccipital |
| **sq** | squamosal |
| **stc** | sella turcica |
| **utf** | upper temporal fenestra |
| **vCL** | lateral head vein (=vena capitis lateralis) |
| **ve** | vestibule |

## ACKNOWLEDGEMENTS

Daniel Snitting and Sifra Bijl (Uppsala University) are thanked for technical support, especially with MIMICS and VGStudio MAX. Feiko Miedema (Paläontologisches Institut, Universität Zürich, PIMUZ) is thanked for discussing aspects of reptilian braincase morphology. Stephan Spiekman (SMNS) is thanked for additional images of *Simosaurus* specimens. Access to specimens were kindly granted by Rainer Schoch (SMNS) and Ingmar Werneburg (GPIT), and the CT datasets were acquired by Tobias Reich (formerly PIMUZ). Thanks to Remí Allemand (MNHN, Muséum national d'Histoire naturelle, Paris, France) for providing images of the cranial endocast of *Libonectes*. We also thank the staff working at the AST-RX facility at MNHN Paris, that made the scanning of the *Simosaurus* specimens possible.

### Funding

The study was supported by the Swiss National Science Foundation (grant nos. 31003A_149506/1 & 31003A_173173 to Torsten M. Scheyer). The funders had no role in study design, data collection and analysis, decision to publish, or preparation of the manuscript.

### Grant Disclosures

The following grant information was disclosed by the authors:
Swiss National Science Foundation: 31003A_149506/1, 31003A_173173.

### Competing Interests

The authors declare there are no competing interests.

### Author Contributions

- Elisa H. London performed the experiments, analyzed the data, prepared figures and/or tables, authored or reviewed drafts of the article, and approved the final draft.
- Dennis F.A.E. Voeten conceived and designed the experiments, analyzed the data, prepared figures and/or tables, authored or reviewed drafts of the article, and approved the final draft.
- Henning Blom conceived and designed the experiments, authored or reviewed drafts of the article, and approved the final draft.
- Torsten M. Scheyer conceived and designed the experiments, performed the experiments, analyzed the data, prepared figures and/or tables, authored or reviewed drafts of the article, and approved the final draft.

### Data Availability

Raw data are available in the Supplementary Figures.
The fossil described herein is officially SMNS: SMNS 16363.

The micro-CT scan data and the models (as PLY/STL) are available at: https://www.morphosource.org/projects/000694362?locale=en [Project ID: 000694362].

## Supplemental Information

Supplemental information for this article can be found online at http://dx.doi.org/10.7717/peerj.19932#supplemental-information.

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
