# Peer review of "The braincase anatomy of *Simosaurus gaillardoti* (Diapsida: Sauropterygia) revealed with X-ray micro-computed tomography"

_PeerJ, doi:10.7717/peerj.19932_

## Round 0.1 · original submission · Major Revisions

Congratulations on your good work. The reviewers have provided comprehensive comments.

Reviewer 1 ·

Basic reporting

-

Experimental design

-

Validity of the findings

-

Additional comments

I would like to congratulate the authors for the excellent job on CT scanning Simosaurus and its interpretation. The manuscript is well-structured and well-written. Please see below for just some minor comments:

l. 48-50: I think both Wang et al. (2022) and Kear et al. (2024) did not specifically address the earliest appearance of sauropterygians. Kear et al. (2023) addressed this but did not provide any concrete evidence of an end-Permian fossil sauropterygian! I suggest that the authors tone down this opinion, avoiding possible confusion that there is indeed an end-Permian fossil sauropterygian.
l. 55: change ‘analysesas’ to ‘analyses as’.
l. 57: add ‘and’ behind ‘environments.
l. 95: note that a recently expanded data matrix of eosauropterygians (Hu et al., 2024, Swiss Journal of Palaeontology) also recovered a sister relationship between Simosaurus and Germanosaurus.
l. change ‘This variation, is for example’ to ‘This variation is, for example’.
l. 104: change ‘Fig. 1, 2’ to ‘Figs. 1, 2’.
l. 142: Is it possible to make this done (detailed renderings of the braincase bones of Simosaurus) in this manuscript, to facilitate the comparison of the braincase between Simosaurus and Nothosaurus?
l. 534: delete the extra comma.
l. 616: It seems that the definition of nothosaurs here is different from elsewhere in the manuscript. The authors may want to clarify it.

·

Basic reporting

Dear Authors,

The authors reconstructed the neuroanatomy of a sauropterygian and reinterpreted some of its traits. The manuscript is well written, although there are some mistakes in the text, like typos or other misspelled words and names that I noticed and marked in the annotated PDF. The comparisons and references are up-to-date and relevant to the topic, the figures are well-prepared (although I have some criticisms, see below), and the manuscript is self-contained. However, I did not see in the manuscript any mention of data availability, either to the 3D models or the tomographic datasets. I understand that sometimes there might be some issues with sharing the CT data (although this would be the ideal posture of any publication based on such data), but there is no reason not to share the 3D models of the segmented regions of interest. It was already challenging not having such models during the review because I could not confirm some of their observations, but if this data is not shared with the publication, then there is a bigger problem, as their observations cannot be reproduced. So I strongly recommend that the authors share their data.

I provide minor comments on the annotated PDF, and the most important issues I highlight below in the next sections.

Regarding the figures, I strongly suggest the authors make a more extensive labelling, which would aid the reader a lot in following their descriptions and interpretations. Some examples of what I mean:

- l. 385: Posteriorly on the ventral surface of the pterygoid, a foramen is
385 present, identified as the likely entry of an artery into the pterygoid, bu
Is this indicated in any of the figures? Is it the aPT?

- The paracondylar interstices are indicated in Figure 3, but it would be very helpful to have this also indicated in Figures 5 and 6; it is quite difficult to see what you mean in 3F. Particularly in Fig.5, where the 3D model of the interstices is absent, it would be nice to have it indicated with a dotted line (like the CN V is indicated in Fig.5F)

- L. 463-464: In Simosaurus, the fenestra ovalis is located between the opisthotic and prootic and is visible in lateral view
This could be shown in Figure 5 as well.

- L. 560-564: This portion is quite difficult to follow. It would be much better if you made reference to the figures and also in the figures identified the foramina by their names, and not only by the structure piercing them

Experimental design

The methods and description of the study design are adequate.

Validity of the findings

I do agree with most of the authors' interpretations of the basicranial and neuroanatomy, but I have two points I wish to highlight.

1. On line 532, you mention the internal carotid artery enters the skull through the jugular foramen, and on the next line that it splits more anteriorly in Simosaurus than in Nothosaurus. However, according to your reconstructions, the jugular foramen has no connection to the model of the internal carotid (and I cannot remember it entering through the jugular foramen in any taxon). Did you mean the cerebral/jugular vein? This is usually the vessel passing through the jugular foramen (Rieppel 1985, Gaffney 1972).

2. The hypoglossal and jugular foramina
Between lines 565-570 and 576-578, you dispute Rieppel (1994)'s claims about the presence of two hypoglossal foramina on the exoccipital of Simosaurus and Nothosaurus. From the description between lines 565-570, I do not think you are discussing the same trait. You mention a larger anterior and a smaller posterior foramen as described by Rieppel as the hypoglossal f. and then proceed to state that your model shows only one opening that corresponds to the jugular foramen. In Rieppel 1994, this would be a third foramen that, together with the foramen magnum, borders a pillar-like structure formed by the exoccipital (Rieppel 1994, p. 12). This pillar-like structure can be seen in your models on Fig. 5H, where it is also possible to see the jugular foramen lateral to it (again, it would be nice to indicate them in the figure). The two hypoglossal foramina (one large and anterior and another smaller and posterior) should pierce this pillar-like structure. Particularly important to consider here is that the absence of a structure in models or on CT data is not strong evidence of the actual absence of such a structure, because there are many factors that could make this structure not be apparent, such as the low resolution, poor contrast, and the segmentation process itself. The hypoglossal foramina tend to be diminutive, and it might just be the case that you were not able to find it in the images. So, I suggest double-checking that in the CT images, and even if you still cannot find them, I would be more cautious with the interpretation.

Additional comments

I hope the authors find my comments and suggestions helpful to improve the quality of the manuscript, and they are invited to reach me if something is not clear in my assessment.

All the best regards,
Gabriel Ferreira

·

Basic reporting

This is a high-quality manuscript, well-sourced and well-assembled. Soft tissue data is excellent. I have suggestions below for how to increase the conviction of the similarity to nothosauroids the authors argue.

Experimental design

Thorough description and figuring of important and previously inacccessible anatomy.

Validity of the findings

I strongly recommend the authors add more to their manuscript by including a figure of a nothosauroid to showcase the anatomical similarities they argue for. Right now, they state several times that the anatomy of Simosaurus resembles a nothosauroid rather than a pistosauroid, or some other sauropterygian. As this part of the tree is currently in flux, the longevity of this contribution would greatly increase if another taxon or two were figured, if only for comparative purposes.

Alternatively, the authors could include Simosaurus and their impressive novel data in one or more recent phylogenies to demonstrate its shared character states with nothosauroids.

In either case, the interesting comments on feeding ecology fall somewhat flat if the reader cannot clearly see that Simosaurus is likely itself a nothosauroid.

Additional comments

Line edits:
Line 48: Please change 252 to ~250, as there are no known Induan sauropterygians

Line 50: The initial diversification of marine reptiles (whether ichthyosaurs+ and sauropterygians are a clade or not) is a hotbed of research currently. Perhaps give us a sentence more on what exactly these authors found to suggest a Permian appearance.

Line 66: Dates given for the Olenekian stage are incorrect

Line 97: Please comment a bit more on the lack of other nothosaurs being included in Simosaurus phylogenetic placements. Why is this important? It will help the reader to sort out the confusing history.

Line 104: plural of fenestra as fenestrae

Lines 127-131: Please give us an idea of what kind of sauropterygians these taxa are, and order them accordingly

Line 239: For the Osteological aspects, please add subheadings to the section to help the reader.

Line 308: This might be a simple preference, but list Milleropsis before Youngina in the three-taxon comparison (with Massospondylus)

Line 543: Is this prootic ‘fenestra’ simply part of the anterior inferior process of the prootic seen in many saurian (crown) reptiles? Please label it in Figure 5 alongside CN V.

---

## Round 0.2 · Minor Revisions

This is a much improved version of the manuscript, however one of the reviewers requires that you respond to one of their concerns that was not adequately addressed.

Reviewer 1 ·

Basic reporting

-

Experimental design

-

Validity of the findings

-

Additional comments

The revised manuscript is much improved after the authors incorporated most of the reviewers' comments into consideration, but there is still one point from my original comments to be revised, as below.

My original comment: l. 48-50: I think both Wang et al. (2022) and Kear et al. (2024) did not specifically address the earliest appearance of sauropterygians. Kear et al. (2023) addressed this but did not provide any concrete evidence of an end-Permian fossil sauropterygian! I suggest the authors tone down this opinion, avoiding possible confusion that there is indeed an end-Permian fossil sauropterygian.

Author response: Thanks, we have now revised the text accordingly.

Author revised text: Although fossils are currently absent from the Latest Permian to the Earliest Triassic (i.e., pre-Spathian), recent studies have discussed the possibility of an earlier appearance of Mesozoic marine reptile clades, including the sauropterygians, just prior to the Permian-Triassic Mass Extinction Event (PTME; Wang et al., 2022; Kear et al., 2023) based on tip-dating and clade-divergence timing estimates (but see also Motani et al., 2017).

New comment: This revision is even worse. First, as far as I known, Wang et al. (2022) did not discuss the possibility of a Permian sauropterygian, but Kear et al. (2023) did. Second, both Wang et al. (2022) and Kear et al. (2023) did not perform tip-dating and clade-divergence timing estimates. Only Motani et al. (2017) did! Please correct me if my memory is wrong.

·

Basic reporting

Well done

Experimental design

Suitable for publication

Validity of the findings

The additions to the MS and supplement meet the recommendations previously laid out.

Additional comments

I appreciate the additions and modifications to the main text and supplement made by the authors. They adequately responded to the revisions recommended by me and the other reviewers, and this study will be an important contribution to the study of Sauropterygia.

---

## Round 0.3 · accepted · Accept

I am pleased to see that all of the suggestions made during the last set of reviews have been taken into account. I am recommending that this manuscript be accepted for publication by PeerJ.

Reviewer 1 ·

Basic reporting

no comment

Experimental design

no comment

Validity of the findings

no comment

Additional comments

The authors have addressed my last piece of concern and corrected my memory. The manuscript could be published in its current form. I will be happy to see its final publication.